# MFF-dependent mitochondrial fission regulates presynaptic release and axon branching by limiting axonal mitochondria size

Tommy L. Lewis Jr [1,2,5], Seok-Kyu Kwon[1,2,6], Annie Lee[1,2], Reuben Shaw[3] & Franck Polleux [1,2,4]

Neurons display extreme degrees of polarization, including compartment-specific organelle morphology. In cortical, long-range projecting, pyramidal neurons (PNs), dendritic mitochondria are long and tubular whereas axonal mitochondria display uniformly short length. Here we explored the functional significance of maintaining small mitochondria for axonal development in vitro and in vivo. We report that the Drp1 'receptor' Mitochondrial fission factor (MFF) is required for determining the size of mitochondria entering the axon and then for maintenance of their size along the distal portions of the axon without affecting their trafficking properties, presynaptic capture, membrane potential or ability to generate ATP. Strikingly, this increase in presynaptic mitochondrial size upon MFF downregulation augments their capacity for $Ca^{2+}$ ($[Ca^{2+}]_m$) uptake during neurotransmission, leading to reduced presynaptic $[Ca^{2+}]_c$ accumulation, decreased presynaptic release and terminal axon branching. Our results uncover a novel mechanism controlling neurotransmitter release and axon branching through fission-dependent regulation of presynaptic mitochondrial size.

[1] Department of Neuroscience, Columbia University, New York, NY 10032, USA. [2] Mortimer B. Zuckerman Mind Brain Behavior Institute, New York, NY 10032, USA. [3] Molecular and Cell Biology Laboratory, Salk Institute for Biological Studies, La Jolla, CA 92037, USA. [4] Kavli Institute for Brain Science at Columbia University, New York, NY 10032, USA. [5] Present address: Aging & Metabolism Research Program, Oklahoma Medical Research Foundation, Oklahoma City, OK 73104, USA. [6] Present address: Center for Functional Connectomics, Brain Science Institute, Korea Institute of Science and Technology, Seoul 02792, South Korea. These authors contributed equally: Tommy L. Lewis Jr, Seok-Kyu Kwon. Correspondence and requests for materials should be addressed to F.P. (email: fp2304@cumc.columbia.edu)

Neurons are among the most highly polarized cells found in nature. This high level of polarization underlies the formation of structural compartments such as the axon, dendrites and spines which in turn dictates information processing and transfer in neural circuits. In order to set-up this extreme degree of compartmentalization and maintain it throughout the life of the organism, critical cellular and molecular effectors of polarization, including mRNAs, proteins, and organelles, must be either differentially trafficked or functionally regulated in a spatially precise way[1].

Mitochondria are one of the most abundant organelles found in neurons and are localized throughout the axonal and somatodendritic domains[2–4]. Several studies have observed that dendritic mitochondria have a long and tubular shape, while axonal mitochondria appear to be short and punctate[2–6]. However, the functional implications of this structural difference have not been addressed. In general, mitochondria play many important physiological functions such as ATP production via oxidative phosphorylation, $Ca^{2+}$ uptake, lipid biogenesis, and can trigger apoptosis through cytochrome-c release[7]. Dendritic mitochondrial morphology and function have mainly been studied in the context of neurodegenerative diseases and their contribution to activity-dependent homeostasis[3,8]. In axons, the presynaptic capture of mitochondria is both necessary and sufficient for terminal axon branching in cortical neurons[9] and these pre-synaptically captured mitochondria can contribute to ATP generation and play an essential role in buffering cytoplasmic $Ca^{2+}$ during synaptic transmission[10–12]. Interestingly, presynaptic $Ca^{2+}$ clearance by individual mitochondria underlies bouton-specific regulation of presynaptic vesicle release along cortical and hippocampal axons[11,12]. Mitochondrial function has been suggested to differ in specific neuronal compartments[3,13], but the outcome of disrupting mitochondrial size in a compartment-specific manner remains to be determined.

Mitochondrial size is regulated through the competing processes of fission and fusion[14]. Mitochondrial fusion is regulated by two distinct molecular effectors: Mfn1/2 mediates outer membrane fusion[15] while Opa1 regulates inner membrane fusion[16]. Fission occurs by the "pinching" of a mitochondrion into two new mitochondria via oligomerization of Drp1, a dynamin-like GTPase, at the outer membrane[17]. Because Drp1 is a cytoplasmic protein, it must be recruited to the outer mitochondrial membrane via Drp1 "receptors." There are currently four known Drp1 receptors: MFF, FIS1, MiD49 (Scmr7), and MiD51 (Scmr7L), whose relative contribution to Drp1-dependent fission is still debated and likely cell-type specific[18–24].

Our understanding of the relative contribution of fission and fusion for neuronal morphogenesis or synaptic function remains fragmented. Loss of Drp1 is embryonic lethal and leads to defects in brain development, neuronal process outgrowth, and synapse formation. However, Drp1 affects many cellular processes other than mitochondrial fission and also severely affects neuronal viability, so this observation does not unequivocally implicate mitochondrial fission as the cause of the defects in neuronal development[25–30]. Mitochondrial fusion is also required for proper development as each of the Mfn1, Mfn2, and Opa1 knockouts are lethal[15,31,32], however their role in cortical neuron development is less well studied.

We establish that the small mitochondrial size characterizing the axon of cortical PNs is dependent on mitochondrial fission factor (MFF), as downregulation of *Mff* results in a striking increase of the size of mitochondria entering the axon, and an inversion of the fission/fusion ratio along the axon. This elongation of mitochondria is not observed in dendrites, and does not affect trafficking along axons, presynaptic localization, mitochondrial membrane potential, or the capacity to generate ATP.

However, the Mff-dependent increase in presynaptic mitochondrial length significantly increased their total $Ca^{2+}$ uptake capacity, resulting in decreased evoked neurotransmitter release and decreased terminal axon branching in vivo. Our results identify a novel molecular mechanism limiting axonal mitochondria size and demonstrate its functional importance for neurotransmitter release and axon branching by limiting the $Ca^{2+}$ buffering capacity of presynaptic mitochondria.

## Results

**Mitochondrial morphology is distinct in axons and dendrites**. In order to visualize mitochondrial morphology and function in developing and adult cortical neurons in vitro and in vivo, we implemented a strategy for sparse labeling via ex utero or in utero electroporation (EUE and IUE, respectively)[33]. We developed Flp recombinase-dependent plasmids expressing either a cytoplasmic fluorescent protein (tdTomato (red), EGFP (green) or mTAGBFP2 (blue)), or a mitochondrial matrix-targeted fluorescent protein (mt-YFP, mt-DsRED), based on a previous report[34]. When co-electroporated with a low concentration of Flp-e plasmid at Embryonic Day (E)15.5, this strategy resulted in sparse labeling of layer 2/3 cortical PNs[9,35,36], and allowed us to visualize mitochondrial morphology at high resolution in single, optically-isolated neurons both in vitro (with EUE) and in vivo (with IUE). Following EUE at E15.5, we acutely dissociated and cultured cortical PNs at high density for three weeks (21 Days In Vitro, (DIV)) and observed striking differences in mitochondrial morphology and occupancy between axons and dendrites (Fig. 1a–c). In dendrites, mitochondria were elongated (0.52 to 8.88 μm (10–90th percentile)) and occupied the majority of proximal and distal dendritic processes (69.6 ± 2.54%), while throughout the axon, mitochondria are short, relatively standard in length (0.3 to 1.08 μm (10–90th percentile)) and occupied only an average of 4.95 ( ± 0.4%) of the axonal length.

To confirm the validity of this observation in vivo, we also performed IUE at E15.5 and visualized sparsely labeled layer 2/3 cortical neurons at postnatal day (P)21. Remarkably, quantitative measurement of mitochondrial size and occupancy were highly conserved in vivo (Fig. 1d–f) where dendritic mitochondria measured 1.31 to 13.28 μm (10–90th percentile) in length and occupied 69.58 ± 2.23% of dendritic processes, while axonal mitochondria measured 0.45 to 1.13 μm (10–90th percentile) and occupied 8.41 ± 0.75% of axon length (Fig. 1g, h). These observations confirm that mitochondrial morphology is strikingly different in the axonal and dendritic compartments of cortical layer 2/3 PNs both in vitro and in vivo.

**Axonal mitochondria size is regulated by MFF**. Based on publicly available RNA-seq data from the cerebral cortex and hippocampus[37,38], Mitochondrial fission factor (*Mff*) is the most abundant of the four mediators of mitochondrial fission (so called Drp1 "receptors") in excitatory PNs. To determine if MFF-mediated fission is required for the regulation of mitochondrial size in the axon, we first validated knockdown constructs for mouse *Mff* via western blot and immunocytochemistry (Supplementary Fig. 1). Following in utero or ex utero electroporation of a 1:1 mixture of the validated shRNA constructs for *Mff* (539 & 665), along with plasmid DNA encoding cytoplasmic filler (tdTomato) and a mitochondria-targeted fluorescent protein (mt-YFP) at E15.5 and visualization of neurons at P21 or 21DIV, mitochondrial length and occupancy along the axon were dramatically increased compared to control both in vivo and in vitro (Fig. 2a–d and Fig. 2i–n). This increase in mitochondrial length and occupancy is rescued by the co-expression of a cDNA plasmid encoding human MFF, which is impervious to the shRNA directed against mouse *Mff* (Fig. 2e–h and Fig. 2o–r),

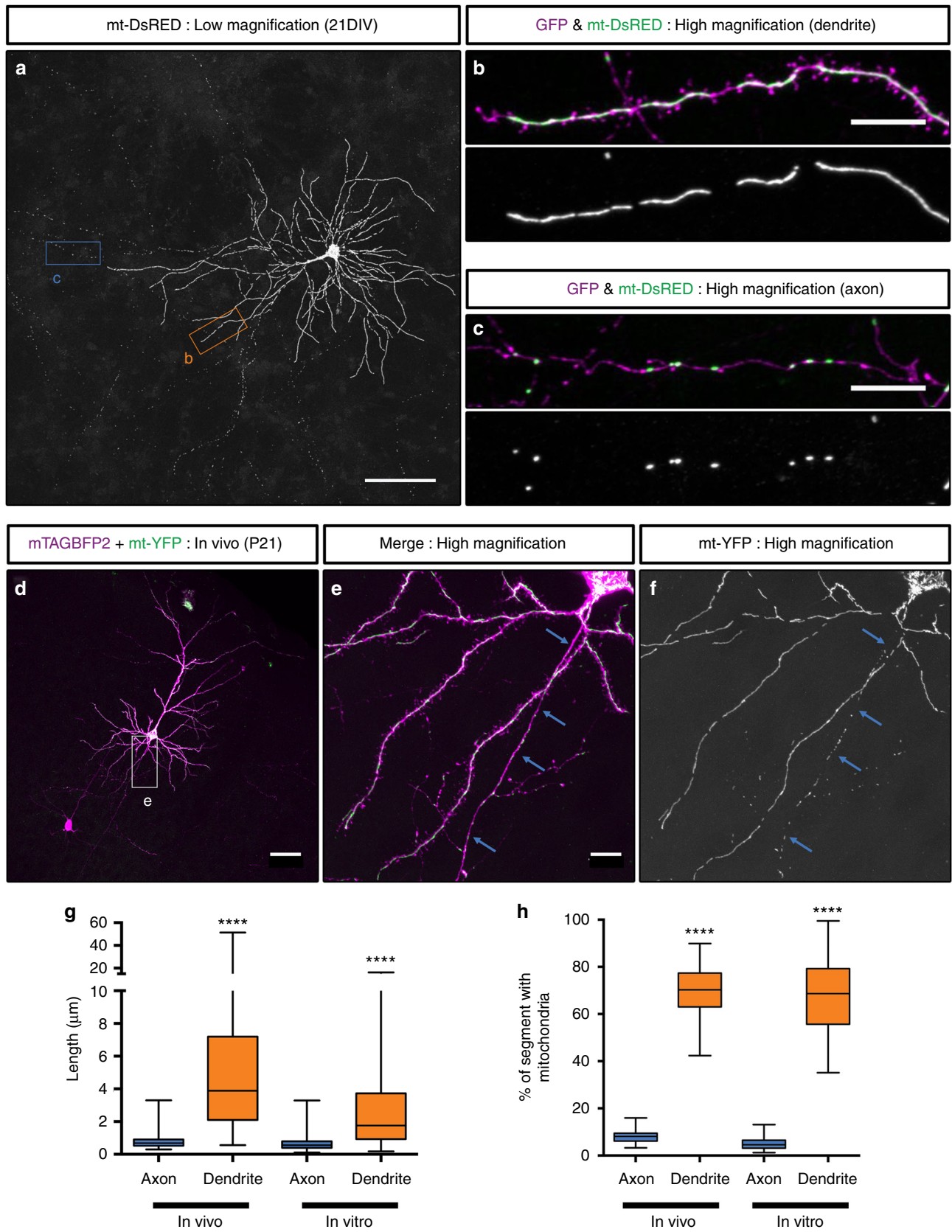

demonstrating that this effect is not the result of an "off-target" effect of the shRNA. Interestingly, *Mff* knockdown had no effect on mitochondrial length or occupancy in dendrites in vivo suggesting a low level of basal MFF activity in this compartment (Supplementary Fig. 2). These results establish that MFF is required for the maintenance of small mitochondrial size in the axon. Interestingly, MFF is present on both somatodendritic and axonal mitochondria (Supplementary Fig. 1) suggesting that fission is not simply regulated by localization of MFF to a particular set of mitochondria (See Discussion).

**Fig. 1** Mitochondria morphology is compartmentally regulated in cortical pyramidal neurons in vitro and in vivo. **a** Mitochondria morphology visualized by cortical ex utero electroporation of a genetically encoded, mitochondria matrix-targeted DsRED (mt-DsRED) and cytoplasmic GFP in layer 2/3 pyramidal neurons at 21 days in vitro (DIV). **b** High magnification of the boxed dendritic region labeled b in **a**. **c** High magnification of the boxed axonal region labeled c in **a**. Notice the striking difference in morphology between elongated, fused mitochondria in dendrites and short mitochondria in the axon. **d** Layer 2/3 cortical neurons at (P)ostnatal day 21 were sparsely labeled in vivo via in utero electroporation with FRT-STOP-FRT plasmids encoding matrix-targeted YFP (mt-YFP) and cytoplasmic filler mTAGBFP2. Sparseness was controlled by lowering the concentration of a Flp-e encoding plasmid until single cell resolution was achieved. **e** High magnification of the boxed region in **d**. **f** The mt-YFP channel of the boxed region of **d**. **g** Quantification of mitochondrial length in the axons and dendrites of L2/3 cortical neurons both in vivo and in vitro, demonstrating that mitochondria in the axon are much shorter than mitochondria in the dendrites. **h** Quantification of the percent of the axonal or dendritic segment occupied by mitochondria both in vivo and in vitro, demonstrating that mitochondria occupy a much larger percentage of the dendritic arbor than of the axon in L2/3 cortical neurons. Data is represented by box plots displaying minimum to maximum values, with the box denoting 25th, 50th (median), and 75th percentile. $n_{axons}$ in vivo = 21 axons, 235 mitochondria; $n_{dendrites}$ in vivo = 26 dendrites, 267 mitochondria; $n_{axons}$ in vitro = 37 axons, 240 mitochondria; $n_{dendrites}$ in vitro = 43 dendrites, 100 mitochondria. Kruskal–Wallis test with Dunn's multiple comparisons test, $p < 0.0001$ for both length and occupancy in vitro and in vivo. Scale bars represent the following lengths: **a** 100 µm, **b–c**, **e** 10 µm, **d** 50 µm

**MFF activity is important for axonal size upon entry**. To better characterize the role of MFF in regulating axonal mitochondrial size, we used the mitochondrial matrix-targeted, photoconvertible fluorescent protein mEos2[39,40], which allowed us to quantify the relative frequency of mitochondrial fission and fusion in axons of cortical PNs. We employed two distinct imaging strategies to determine where MFF activity is required for axonal mitochondrial size maintenance; (1) photo-conversion of mitochondria located in the cell body followed by time-lapse imaging of mitochondrial entry into the axon, or (2) photo-conversion of a small fraction of mitochondria localized along the axon shaft followed by time-lapse imaging and probing the relative frequency of fusion and fission events (Fig. 3a and Supplementary Fig. 3a). Time-lapse imaging of mitochondrial entry into the axon following photo-conversion in the cell body revealed that loss of MFF affects both the frequency of axonal entry as well as their size. On average ~8 mitochondria ($8.1 \pm 1.7$) entered the axon from the cell body per hour in control neurons, but upon *Mff* knockdown this decreased by more than half ($2.8 \pm 0.78$) (Supplementary Fig. 3b–d). Concurrently, the length of mitochondria entering the axon increased fourfold upon *Mff* knockdown compared to control neurons (Supplementary Fig. 3e and Supplementary Movie 1), establishing that MFF activity in the cell body is required for regulation of the size and number of mitochondria entering the axon.

To determine if MFF-dependent fission is also required along the axon for the maintenance of mitochondrial size, we performed time-lapse imaging following photo-conversion of a small subset of mitochondria along the axon shaft. In control axons, the hourly rate of fission ($4.2 \pm 0.53$) and fusion ($3.1 \pm 0.42$) slightly favors fission. However, upon loss of MFF activity, axonal fission is much less likely ($1.8 \pm 0.30$) while fusion is slightly more likely ($4.2 \pm 0.52$) (Fig. 3b–d) leading to a significant > 3-fold reduction in the fission/fusion ratio ($1.5 \pm 0.13$ (control shRNA) vs. $0.46 \pm 0.08$ (*Mff* shRNA) Fig. 3e and Supplementary Movie 2). These results demonstrate that MFF is required both for regulation of mitochondrial size during axonal entry as well as along the axon shaft for maintenance of small mitochondrial size.

**Mff knockdown disrupts terminal axon branching**. To determine the consequence of decreased MFF expression on axonal development, we performed unilateral cortical in utero electroporation (IUE) of the validated *Mff* shRNA constructs along with a plasmid encoding cytoplasmic tdTomato at E15.5, and collected coronal sections at P21 to visualize axonal growth and branching. In control brains, axons grew across the corpus callosum reaching the contralateral hemisphere where they invaded the upper layers forming a stereotyped branching pattern in layers 2/3 and 5

(Fig. 4a–c)[9,35,36]. Knockdown of *Mff* did not affect neuronal migration, axon formation and axon growth across the midline. Axons from *Mff* knockdown neurons reached their target territory on the contralateral cortex all the way to superficial layers 2/3. However, terminal axon branching was dramatically reduced, especially in layers 2/3 (Fig. 4d–f). This striking reduction of cortical axon branching was rescued upon reintroduction of shRNA-impervious human *MFF* validating the specificity of our shRNA directed against *Mff* (Fig. 4g–k). These results emphasize the importance of MFF-dependent fission for the control of axonal mitochondria size and axonal branching in vivo.

**MFF knockdown does not affect presynaptic capture**. Based on our previous work demonstrating the importance of presynaptic mitochondrial capture for terminal axonal branching[9], we tested whether altered mitochondrial size had any effect on axonal mitochondrial transport and/or their capture at presynaptic sites. As previously shown, mitochondrial motility in vitro and in vivo progressively decreases along the axons of PNs[40–42], so we quantified mitochondrial motility at both 7DIV and 21DIV (Supplementary Fig. 4a, b and Supplementary Movie 3). We observed a modest but significant increase in the number of stationary mitochondria at 7DIV in MFF-deficient axons compared to control. Strikingly as seen in Supplementary Movie 3, these elongated axonal mitochondria are clearly capable of sustained trafficking along the axon (Supplementary Fig. 4c, d). At 21DIV, the vast majority of these elongated mitochondria become immobilized at specific points along the axon as observed in control axons ($94.8 \pm 3.53\%$ vs. $91.3 \pm 3.35\%$ in control).

To determine if altering mitochondrial size along the axon affected their capture at presynaptic boutons in vivo, we performed unilateral cortical IUE of the validated shRNA constructs for *Mff* (1:1 of 539:665) along with plasmid DNA encoding cytoplasmic tdTomato (cell filler), mt-YFP (mitochondrial marker) and VGLUT1-HA (presynaptic marker) at E15.5 and collected coronal sections at P21 to visualize co-localization of mitochondria and VGLUT1 positive presynaptic sites. Interestingly, elongation of axonal mitochondrial size had no significant effect on localization to presynaptic sites or the density of VGLUT1 presynaptic boutons along layer 2/3 cortical axons (Supplementary Fig. 4e–h). These results demonstrate that elongation of axonal mitochondria size does not impact their trafficking along cortical axons or their capture at presynaptic boutons and suggested that we should instead focus on potential changes in mitochondrial function that might accompany the observed increase in mitochondrial size upon knockdown of *Mff* to explain its effect on axon branching.

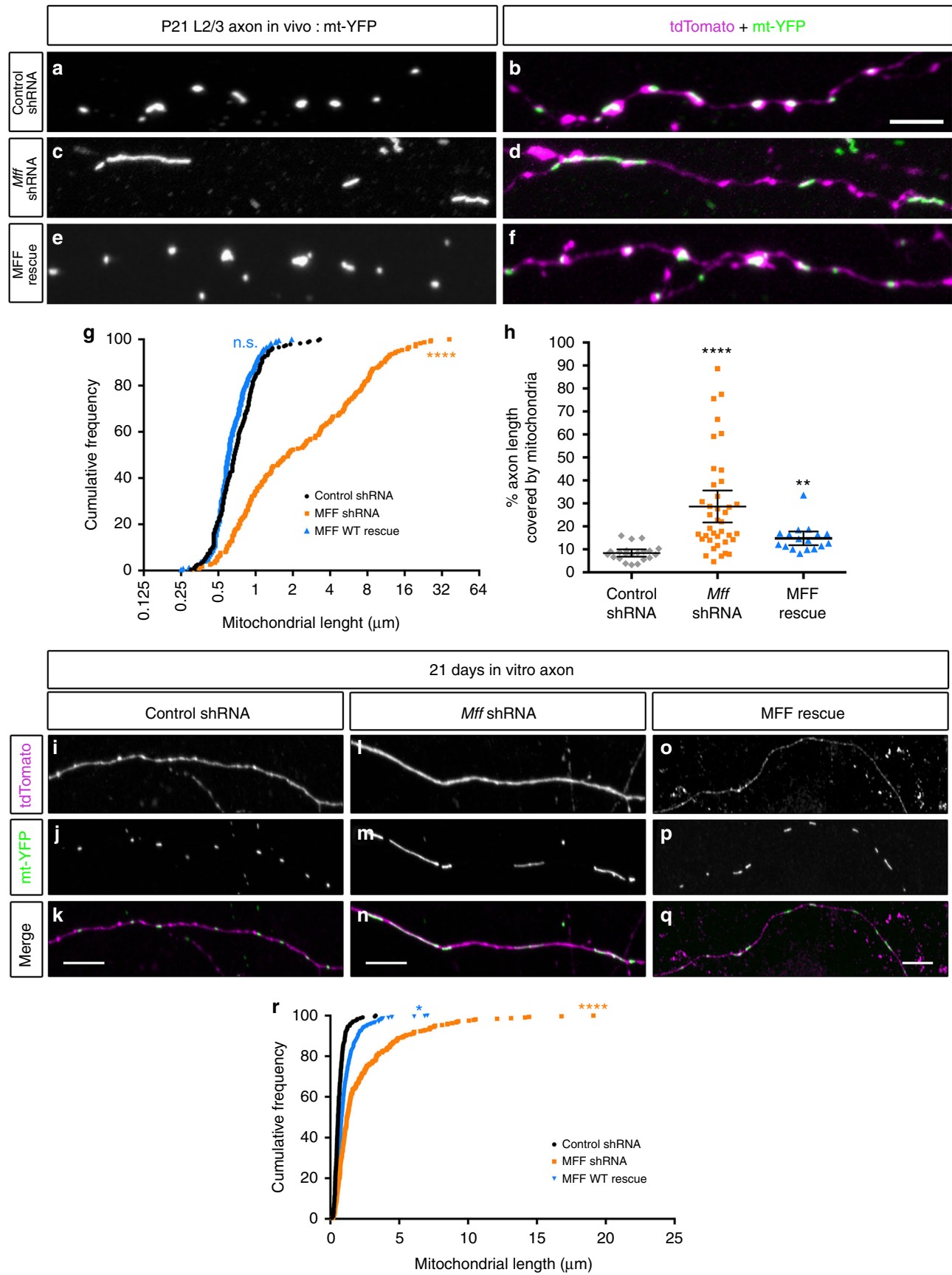

**Fig. 2** Loss of Mitochondrial fission factor (MFF) increases axonal mitochondrial size. **a** Mitochondria morphology visualized in a P21 axon after in utero electroporation of mt-YFP, cytoplasmic tdTomato, empty pCAG vector, and control shRNA. **b** Merge of the mt-YFP and tdTomato channels. **c** Mitochondria morphology visualized as above plus a 1:1 mixture of *Mff* shRNAs (539 and 665). **d** Merge of the mt-YFP and tdTomato channels. **e** Mitochondria morphology visualized as above plus pCAG Flag-hMFF, and a 1:1 mixture of *Mff* shRNAs (539 and 665). **f** Merge of the mt-YFP and tdTomato channels. **g** Cumulative distribution of mitochondria at various sizes under the three conditions above. There is a clear shift to larger mitochondria upon loss of MFF, which can be rescued by reintroduction of a shRNA-impervious human *MFF*, establishing that MFF is the main regulator of mitochondrial size along the axon. **h** Quantification of axonal mitochondrial occupancy showing that loss of MFF activity leads to an increase of axonal occupancy. Data is represented as a scatter plot with mean ± sem. $n_{control\ shRNA}$ = 21 axons, 235 mitochondria; $n_{MFF\ shRNA}$ = 39 axons, 237 mitochondria; $n_{MFF\ rescue}$ = 17 axons, 237 mitochondria. Kruskal–Wallis test with Dunn's multiple comparisons test; length: $p < 0.0001$ for control vs. shRNA and rescue vs. shRNA, $p$ = no significance for control vs. rescue; occupancy: $p < 0.0001$ for control vs. shRNA, $p < 0.01$ for control vs. rescue. **i–k** Representative images of an axon from a neuron electroporated with mt-YFP, tdTomato and control shRNA at E15.5, and visualized at 21DIV. **l–n** Representative images as above but a 1:1 mixture of *Mff* shRNAs (539 and 665). **o–q** Representative images as above but pCAG Flag-hMFF and a 1:1 mixture of *Mff* shRNAs (539 and 665). **r** Cumulative frequency of mitochondrial length for the three conditions above in 21DIV axons. Each quantification is from three independently electroporated pups. $n_{control\ shRNA}$ = 37 axons, 240 mitochondria; $n_{MFF\ shRNA}$ = 31 axons, 216 mitochondria, $n_{MFF\ rescue}$ = 35 axons, 355 mitochondria. Kruskal–Wallis test with Dunn's multiple comparisons test. length: $p < 0.0001$ for control vs. *Mff* shRNA, $p < 0.05$ for control vs. *Mff* rescue. Scale bars represent the following lengths: **b** 5 μm, **k**, **n**, **q** 10 μm

**Reduced MFF activity does not impair mitochondria function.** We first examined if either mitochondrial membrane potential or mitochondrial ATP levels were affected following a reduction in MFF levels and increased mitochondrial size along cortical axons. To determine if these elongated axonal mitochondria maintain their membrane potential, we used Tetramethylrhodamine, Methyl Ester (TMRM, 10 nM). We electroporated cortical neurons with either control shRNA or *Mff* shRNA and unique combinations of mitochondrial and cytoplasmic fluorescent proteins (mt-mTAGBFP2 and Venus for control vs. mt-YFP and mTAGBFP2 for *Mff* knockdown), and mixed control and knockdown neurons at a 1:1 ratio (Supplementary Fig. 5a–d). This approach allowed us to image mitochondrial membrane potential in control and *Mff* knockdown neurons under the exact same culture conditions. Upon quantification of the mitochondrial-to-cytoplasmic fluorescence (Fm/Fc) ratio for individual mitochondria along the axon[43], the longer axonal mitochondria upon *Mff* knockdown actually have an increased ratio (data not shown) but after accounting for mitochondrial area there is no change in the $F_m/F_c$ ratio vs. the control mitochondria ($15.0 ± 2.63$ in *Mff* knockown vs. $13.8 ± 1.08$ for control, Fig. 5a, b). To confirm that our TMRM labeling is sensitive to changes in mitochondrial function, and to determine the portion of membrane potential due to oxidative phosphorylation in control and knockdown 21DIV neurons, we treated the neuron cultures sequentially with Antimycin A (Complex III inhibitor, $1.25\ μM$) and FCCP ($1.25\ μM$) while visualizing mitochondria via mt-mTAGBFP2 and TMRM. Upon Antimycin A addition, we observe a ~30% decrease in TMRM intensity for both control and *Mff*-knockdown mitochondria (Supplementary Fig. 5e, f). These results demonstrate that mitochondria in *Mff*-knockdown neurons are performing oxidative phosphorylation at similar levels to those in control neurons.

Taking a similar approach, we measured intra-mitochondrial ATP levels using a genetically-encoded fluorescence resonance energy transfer (FRET)-based probe, ATEAM1.03, targeted to the mitochondrial matrix[44] (mt-ATEAM and mCardinal for control vs. mt-ATEAM and mScarlet for *Mff* knockdown). Again, we found no significant change in the YFP/CFP ratio for mitochondria upon *Mff* knockdown ($2.55 ± 0.07$) vs. control ($2.38 ± 0.08$) imaged with the same settings and culture conditions at 21DIV (Fig. 5c, d), suggesting that ATP generation capacity of the elongated axonal mitochondria upon *Mff* knockown does not differ from control axonal mitochondria.

Next, we tested if there was a change in the redox potential of elongated, Mff-deficient mitochondria using a genetically-encoded redox-sensitive probe targeted to the mitochondrial

matrix (mt-roGFP2[45]; see validation of the probe for axonal mitochondria in Supplementary Fig. 5g, h). Upon *Mff* knockdown, there was no significant change in fluorescence emission ratio upon the excitation with 405 nm/488 nm lasers ($0.77 ± 0.04$) compared to neurons expressing control shRNA ($0.70 ± 0.04$) at 21DIV (Fig. 5e, f). These results argue that reducing MFF-dependent mitochondrial fission in axons results in significant elongation of axonal mitochondria but does not significantly affect their membrane potential, their ability to generate ATP through oxidative phosphorylation or their redox state.

**Mitochondrial elongation increases mitochondrial Ca$^{2+}$ uptake.** Based on previous results from our lab and others demonstrating that mitochondrial Ca$^{2+}$ uptake plays a critical role in regulating presynaptic release at en passant boutons of cortical PNs[11,12], we tested if mitochondrial Ca$^{2+}$ uptake was affected upon *Mff* knockdown. Our hypothesis was that the total Ca$^{2+}$ uptake capacity of mitochondria is in part dependent on their total matrix volume. To monitor presynaptic mitochondrial Ca$^{2+}$ dynamics ([Ca$^{2+}$]$_m$), we used a genetically-encoded Ca$^{2+}$ sensor targeted to the mitochondrial matrix, mt-GCaMP5G[11], and co-expressed this construct together with VGLUT1-mCherry and mt-mTAGBFP2 (a constitutive mitochondrial matrix label) into cortical PNs using EUE at E15.5. At 17-23DIV, we imaged intra-mitochondrial Ca$^{2+}$ dynamics before and following stimulation of presynaptic release with 20 action potentials (AP) at 10 Hz using a concentric bipolar electrode[11]. During stimulation, Ca$^{2+}$ import occurs at points of these long mitochondria in direct contact with presynaptic sites, followed by diffusion of mitochondrial Ca$^{2+}$ along the mitochondria in *Mff* knockdown neurons (Fig. 6, Supplementary Fig. 6 and Supplementary Movie 4). In addition, long mitochondria show significantly faster extrusion of mitochondrial Ca$^{2+}$ after stimulation compared to control mitochondria ($τ = 13.2 ± 1.5$ s of vs. $8.2 ± 1.1$ s in control, Fig. 6a, b, e). To determine if these elongated mitochondria accumulate more Ca$^{2+}$, we measured the integrated intensity of mt-GCaMP5G (area under the curve i.e., quantal content) over the entire length of individual mitochondria associated with single presynaptic VGLUT1 + boutons. Interestingly, elongated presynaptic mitochondria import significantly higher amounts of Ca$^{2+}$ than punctate mitochondria (as measured by area under the curve or total charge transfer: $2.3 × 10^5 ± 0.41 × 10^5$ vs. $1.4 × 10^5 ± 0.19 × 10^5$ in control, Fig. 6c, d). Finally, the resting [Ca$^{2+}$]$_m$ level of these long mitochondria is higher than control (short) mitochondria (Supplementary Fig. 7a), which simply reflects the increase in mitochondrial size/matrix volume (Supplementary Fig. 7b).

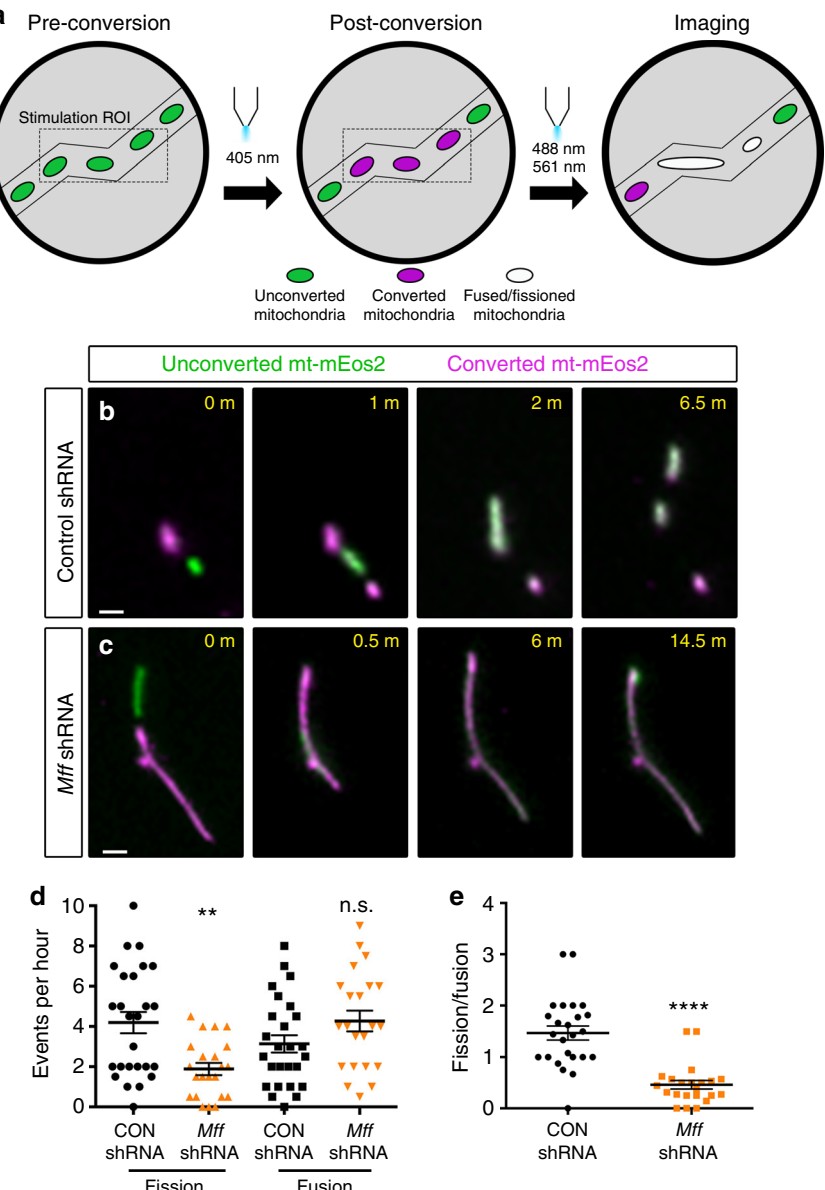

**Fig. 3** Loss of MFF reduces axonal fission and reverses fission/fusion balance. **a** Schematic of the imaging paradigm used for measuring axonal fission and fusion via mitochondria-targeted mEos2 (mt-mEos2), a photoconvertible fluorescent probe. **b** Selected timeframes of a mitochondrial fusion and fission event along the axon of a 7DIV neuron ex utero electroporated with control shRNA and mt-mEos2. **c** Selected timeframes of a mitochondrial fusion event along the axon of a 7DIV neuron ex utero electroporated with a 1:1 mixture of *Mff* shRNA (539:665) and mt-mEos2. Fusion events are mostly coupled with a subsequent fission event in control axons, whereas in MFF-deficient axons, fission is significantly less frequent following a fusion event. See also Supplementary Movie 2. **d** Quantification of the frequency of fission and fusion (#events per hour) per axon segment. Data is represented as a scatter plot with mean ± sem. $p = 0.0019$ for control vs. *Mff* shRNA fission, $p = 0.1135$ for control vs. *Mff* shRNA fusion. **e** Quantification of the fission to fusion ratio per axon segment. Data is represented as a scatter plot with mean ± sem. $p < 0.0001$ for control vs. *Mff* shRNA fission/fusion ratio. Mann–Whitney test. $n_{control\ shRNA} = 26$ axons; $n_{MFF\ shRNA} = 22$ axons. In both **d** and **e**, each dot is a single axon segment. Quantification from three independent cultures in each condition. Scale bars represent the following lengths: 1 μm

To test total $Ca^{2+}$ transferred to elongated mitochondria following stimulation while excluding the impact of fast extrusion, we applied a mitochondrial $Na^+/Ca^{2+}$ exchanger (NCLX) antagonist, CGP 37157, as NCLX is one of the main mitochondrial $Ca^{2+}$ extrusion mechanisms. Interestingly, this inhibitor effectively delays $Ca^{2+}$ extrusion from presynaptic mitochondria and actually increases the difference of total $Ca^{2+}$ charge transfer between *Mff* knockdown and control ($5.9 \times 10^5 \pm 7.7 \times 10^4$ vs. $2.2 \times 10^5 \pm 2.1 \times 10^4$ in control, Fig. 6f, g). These results demonstrate that the longer presynaptic mitochondria induced by *Mff* knockdown underlies a significant increase in their $Ca^{2+}$ uptake capacity.

**Increased uptake of $Ca^{2+}$ decreases presynaptic $Ca^{2+}$ levels.** To understand the impact of this increased mitochondrial $Ca^{2+}$ uptake, we tested if presynaptic cytoplasmic $Ca^{2+}$ ($[Ca^{2+}]_{cyto}$) is affected by the increased $Ca^{2+}$ uptake from mitochondria in *Mff* knockdown axons. Combining GCaMP5G targeted to presynaptic boutons (VGLUT1-GCaMP5G) with mt-mTAGBFP2 and VGLUT1-mCherry allowed us to measure presynaptic $Ca^{2+}$ dynamics at presynaptic boutons associated with mitochondria[11]. All three plasmids with either control or a 1:1 mixture of *Mff* shRNA (539:665) were introduced by EUE at E15.5 and imaged in axons of cortical PNs at 17-23DIV using the stimulation

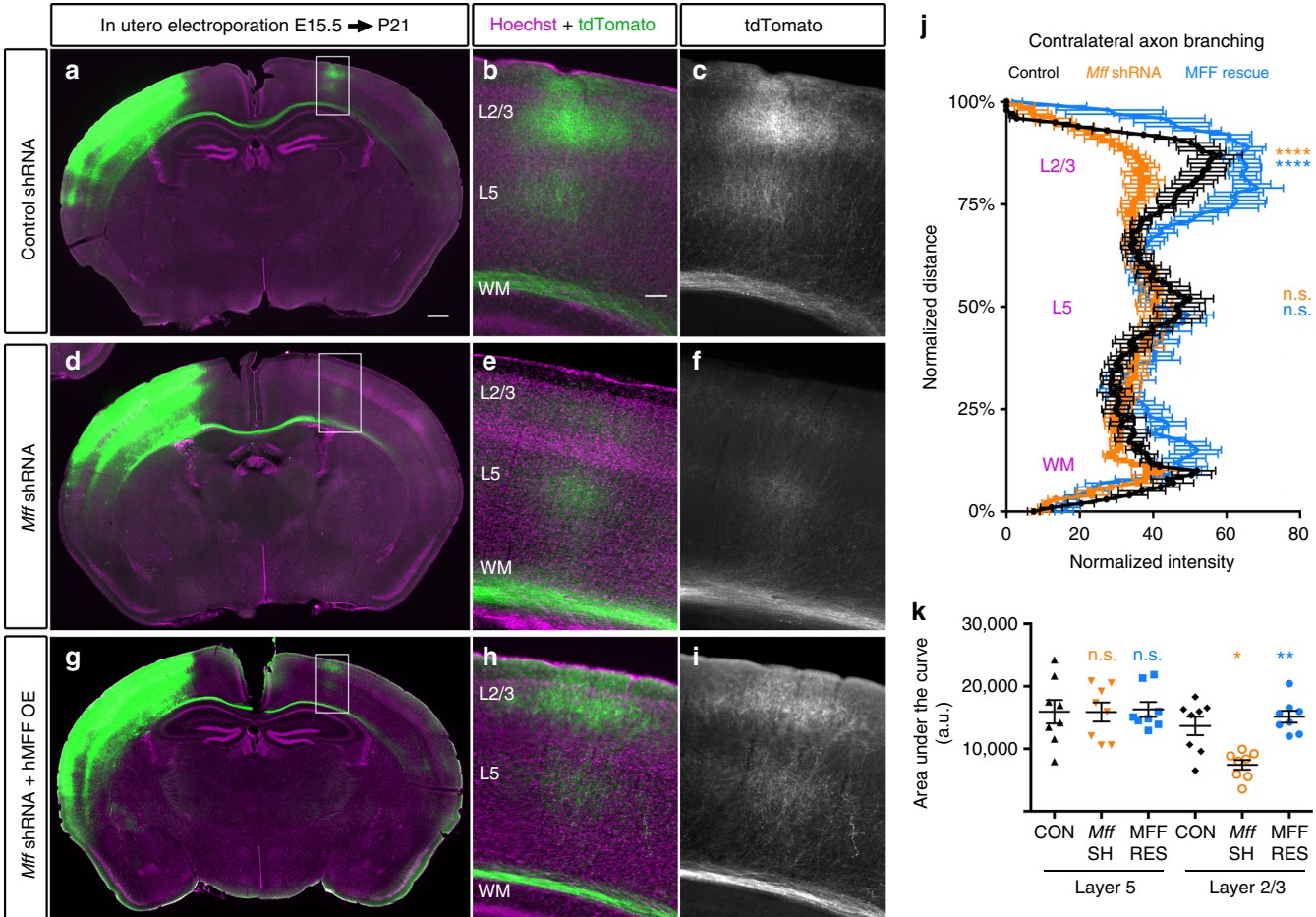

**Fig. 4** Loss of MFF dramatically reduces contralateral axon branching. **a** Low magnification of a coronal section from a P21 mouse brain following cortical IUE with tdTomato, empty pCAG vector and control shRNA. **b** Higher magnification of the box in **a**. **c** The tdTomato channel only from **b** showing the axon branching pattern of contralateral layer 2/3 cortical axons, which branch densely in layer 2/3 and layer 5 but avoid branching in layers 4 and 6. **d** Low magnification of a coronal section from a P21 mouse brain following IUE with tdTomato, empty pCAG vector and a 1:1 mixture of *Mff* shRNAs (539 and 665; see Supplementary Fig. 1 for validation). **e** Higher magnification of the box in **d**. **f** The tdTomato channel only from **e** showing significantly reduced branching in layer 2/3 of the contralateral hemisphere. **g** Low magnification of a coronal section from a P21 mouse brain following cortical IUE with tdTomato, pCAG HA-hMFF (rescue construct) and the 1:1 mixture of *Mff* shRNA (539:665). **h** Higher magnification of the box in **g**. **i** The tdTomato channel only from **h** showing a rescue of layer 2/3 branching upon reintroduction of MFF. **j** Quantification of tdTomato fluorescence along the radial axis of the contralateral cortex (mean ± sem, two-way analysis of variance (ANOVA)) confirming decreased contralateral terminal branching upon MFF loss, and rescue upon reintroduction of shRNA-impervious hMFF. ****$p < 0.0001$ and n.s., not statistically significant. **k** Quantification of the area under the curves shown in **j**. Data is represented as a scatter plot with mean ± sem. $N = 4$ brains for each condition. Each dot represents a single section analyzed. Kruskal–Wallis test with Dunn's multiple comparisons test. n.s., $p > 0.05$; *$p < 0.05$; **$p < 0.01$, and ****$p < 0.0001$. In **j** and **k**, orange characters are for statistical comparisons between control and *Mff* shRNA; blue characters are for statistical comparisons between control and *Mff* shRNA + h*MFF* cDNA rescue condition. Scale bars represent the following lengths: **a** 500 μm, **b** 100 μm

protocol described above (20APs at 10 Hz). Interestingly, the peak of $[Ca^{2+}]_{cyto}$ signals from long mitochondria-associated presynaptic boutons was 20% lower ($0.46 \pm 0.025$ vs. $0.56 \pm 0.02$ in control, Fig. 7a, b). In addition, both the peak values of presynaptic $[Ca^{2+}]_{cyto}$ and the total charge transfer (area under the curve) were significantly reduced in *Mff* knockdown neurons (Fig. 7c–e). In MFF-deficient axons, presynaptic boutons associated with elongated mitochondria display lower cytoplasmic $Ca^{2+}$ levels than boutons without elongated mitochondria (Supplementary Fig. 7c). Therefore, the difference between control and *Mff* knockdown is not due to a global or nonspecific effect of *Mff* knockdown. Moreover, acute MCU inhibitor treatment (Ru360) significantly raised cytoplasmic $Ca^{2+}$ levels at presynapses associated with long mitochondria during AP-evoked neurotransmitter release (Supplementary Fig. 7e, f). These results demonstrate that the increased, MCU-dependent, $[Ca^{2+}]_m$

uptake capacity characterizing elongated presynaptic mitochondria upon Mff knockdown significantly impacts presynaptic $[Ca^{2+}]_{cyto}$ accumulation during neurotransmission.

**Presynaptic release is impaired in *Mff* knockdown neurons.** Presynaptic $Ca^{2+}$ influx through voltage-gated $Ca^{2+}$ channels (VGCC) triggers synaptic neurotransmitter release, therefore the amount of presynaptic $Ca^{2+}$ quantitatively regulates neurotransmitter release at individual boutons[11,46]. To monitor presynaptic release at individual boutons, we employed pHluorin-tagged synaptophysin (syp-pHluorin)[47]. In short, pH-sensitive GFP, pHluorin, is fused to the luminal domain of synaptophysin, and syp-pHluorin fluorescence in synaptic vesicles is quenched because of the low/acidic luminal pH of neurotransmitter vesicles. When exposed to extracellular pH following vesicle exocytosis, the

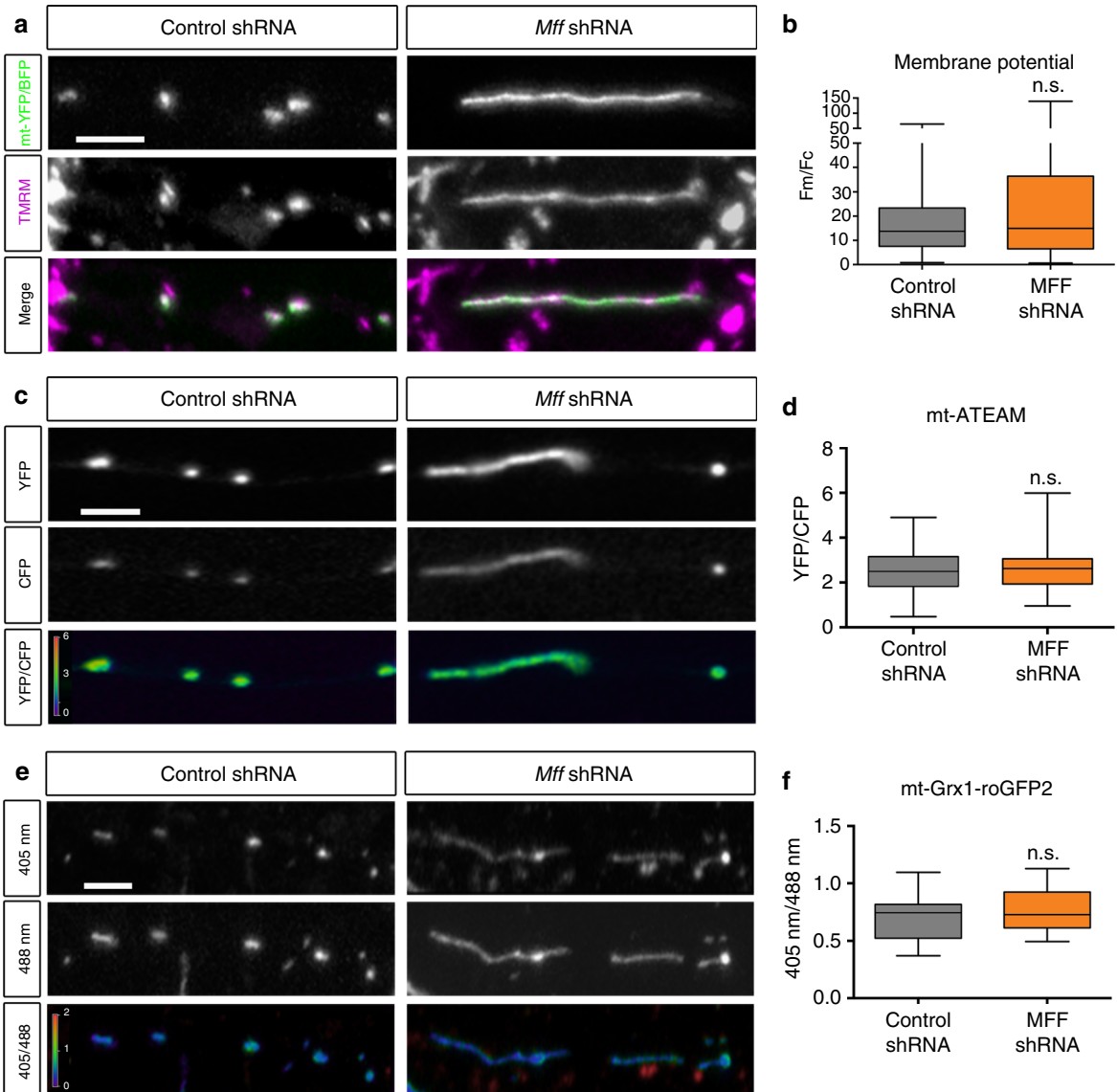

**Fig. 5** Loss of MFF does not significantly alter mitochondrial membrane potential, mitochondrial ATP levels or the redox potential of the mitochondrial matrix. **a** Representative images from axons of neurons electroporated with mt-mTAGBFP2, Venus (cell filler) and control shRNA (left panels) or mt-YFP, mTAGBFP2 (cell filler) and a 1:1 mixture of *Mff* shRNA (539:665) (right panels) via EUE at E15.5, and treated with 10 nM TMRM at 7DIV. **b** Quantification of ($(F_{mitochondria}/F_{cytoplasm})/(Mitochondrial\ Area)$) for TMRM at 7DIV. Data is represented as minimum to maximum box plots, with the box denoting 25th, 50th (median), and 75th percentile. $n_{control\ shRNA} = 7$ axons, 125 mitochondria; $n_{MFF\ shRNA} = 11$ axons, 104 mitochondria, $p = 0.2176$ by non-parametric Mann–Whitney test. **c** Representative images from axons of neurons electroporated with mt-ATEAM, mCardinal and control shRNA (left panels) or mt-ATEAM, mScarlet and 1:1 mixture of *Mff* shRNA (539:665) (right panel) via EUE at E15.5 and visualized at 21DIV. **d** Quantification of ($F_{YFP}/F_{CFP}$) for ATEAM at 21DIV. Data is represented as minimum to maximum box plots, with the box denoting 25th, 50th, and 75th percentile. $n_{control\ shRNA} = 13$ axons, 176 mitochondria; $n_{MFF\ shRNA} = 18$ axons, 145 mitochondria, $p = 0.4504$ by Mann–Whitney. **e** Representative images from axons of neurons electroporated with mt-Grx1-roGFP2 and control shRNA (left panels) or mt-Grx1-roGFP2 and 1:1 mixture of *Mff* shRNA (539:665) (right panels) via cortical EUE at E15.5 and visualized at 21DIV. **f** Quantification of ($F_{405nm}/F_{488nm}$) for roGFP2 at 21DIV. Data is represented as minimum to maximum box plots, with the box denoting 25th, 50th, and 75th percentile. $n_{control\ shRNA} = 28$ axons, 405 mitochondria; $n_{MFF\ shRNA} = 22$ axons, 238 mitochondria, $p = 0.3267$ by Mann–Whitney test. Scale bars represent the following lengths: 5 μm

luminal pH equilibrates closer to pH 7 and pHluorin emits fluorescence. We introduced syp-pHluorin, synaptophysin-mCherry (constitutive presynaptic marker), and mt-mTAGBFP2 with either control or a 1:1 mixture of *Mff* shRNA (539:665) using EUE at E15.5 and imaged at 20-23DIV. Consistent with decreased presynaptic $[Ca^{2+}]_{cyto}$ levels, presynaptic boutons associated with long mitochondria in *Mff* knockdown axons displayed significantly decreased evoked neurotransmitter release ($0.074 \pm 0.012$ vs. $0.13 \pm 0.014$ in control, Fig. 7f–i). In addition, presynaptic boutons associated with elongated mitochondria show lower evoked

neurotransmitter release (syn-pHluorin) than boutons not associated with mitochondria in MFF-deficient axons during evoked neurotransmission (Supplementary Fig. 7d) strongly suggesting that the observed phenotypes are not due to a global, nonspecific effects of *Mff* knockdown but rather due to Mff-dependent increase in mitochondrial size. These results demonstrate that, in cortical axons, regulation of mitochondrial size by MFF-dependent fission is critical for proper presynaptic $Ca^{2+}$ homeostasis, neurotransmitter release and results in decreased terminal axon branching in vivo.

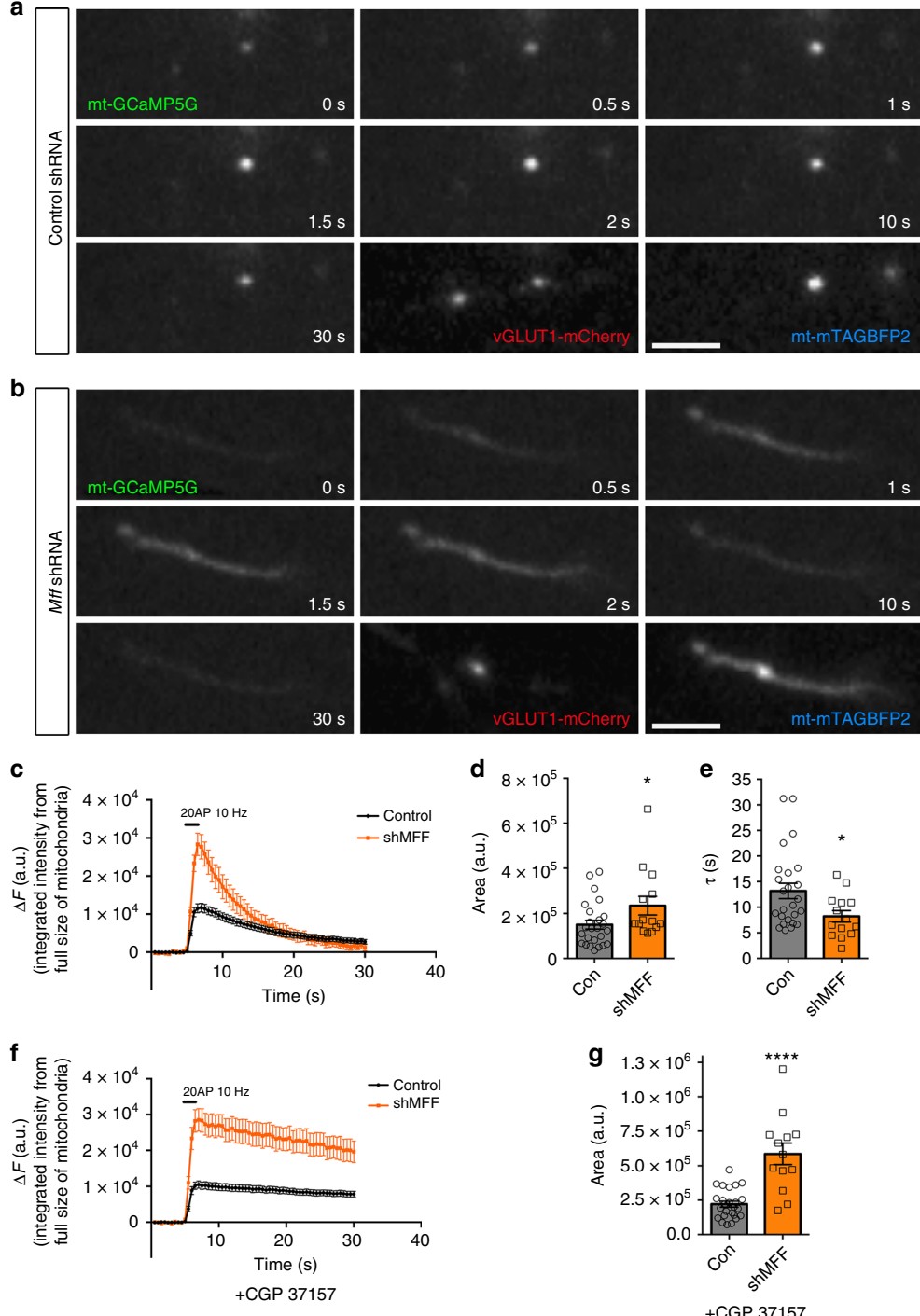

**Fig. 6** Elongated mitochondria in *Mff* knockdown axons uptake more Ca$^{2+}$ upon evoked neurotransmitter release from presynaptic sites. Presynaptic mitochondrial Ca$^{2+}$ was monitored using mitochondria-targeted GCaMP5G (mt-GCaMP5G) with VGLUT1-mCherry and mt-mTAGBFP2 in cortical cultured neurons following EUE at E15.5 and imaged at 17-23DIV. **a**, **b** Cropped time-lapse images of mt-GCaMP5G with repetitive stimulation (20AP at 10 Hz) in control and *Mff* knockdown axons. Mitochondrial Ca$^{2+}$ diffuses along elongated mitochondria from a presynaptic site in an *Mff* knockdown axon (See Supplementary Fig. 6 and Supplementary Movie 4) and extrudes faster than small mitochondria. **c** Integrated intensity of mt-GCaMP5G signals from full-length mitochondria associated with single presynaptic sites is plotted with mean ± sem. **d** Quantification of [Ca$^{2+}$]$_{mt}$ (area under the curve) show long mitochondria with a single presynaptic bouton in *Mff* knockdown axons accumulate significantly more [Ca$^{2+}$]$_{mt}$ than small mitochondria in control axons. *$p < 0.05$, Mann–Whitney test. **e** Long mitochondria show faster decay than control mitochondria. *$p < 0.05$, Mann–Whitney test. **f**-**g** Blocking extrusion of mitochondrial Ca$^{2+}$ by inhibition of mitochondrial Na$^+$/Ca$^{2+}$ exchanger (NCLX) causes more accumulation of [Ca$^{2+}$]$_{mt}$ in long mitochondria. Images were captured after CGP 37157 incubation (10 μM, 3 min) from the same axons in **c**. All bar graphs are represented with mean ± sem. $n_{control} = 7$ dishes, 25 mitochondria; $n_{MFF\ shRNA} = 11$ dishes, 14 mitochondria. ****$p < 0.0001$, Unpaired *t*-test. Scale bars represent the following lengths: 5 μm

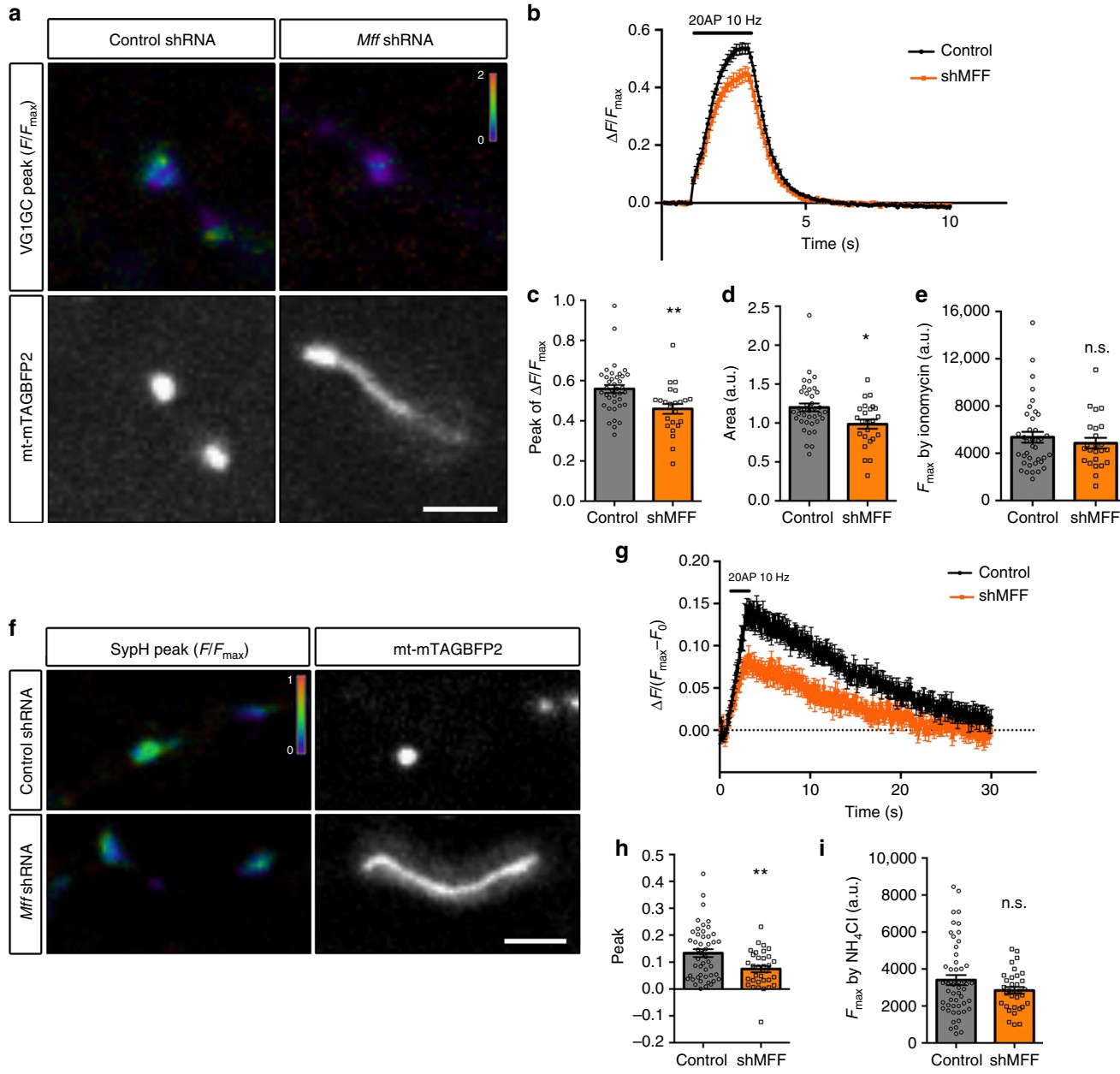

**Fig. 7** Presynaptic boutons associated with long mitochondria in MFF-deficient axons show decreased Ca$^{2+}$ accumulation and reduced evoked neurotransmitter release. **a–e** Presynaptic Ca$^{2+}$ dynamics in *Mff* knockdown cortical neurons were monitored using VGLUT1-GCaMP5G with mt-mTAGBFP2 and VGLUT1-mCherry at 17-23DIV. **a** Representative images show VGLUT1-GCaMP5G peak at 20APs (10 Hz) and mt-mTAGBFP2. VGLUT1-GCaMP5G is displayed by ratio view normalized ($\Delta F/F_{max}$) by $F_{max}$ values obtained following ionomycin (5 μM) treatment at the end of each imaging session. **b–e** Presynaptic boutons from MFF-deficient neurons have significantly decreased peak value and total charge transfer (area under the curve), while $F_{max}$ values are not different. All images were recorded using the same capturing condition. All graphs are represented with mean ± sem. $n_{control}$ = 9 dishes, 38 boutons; $n_{MFF\ shRNA}$ = 11 dishes, 24 boutons. $p$ = 0.001 for peak, $p$ = 0.018 for area, Mann–Whitney test. **f–i** Presynaptic release properties linked to long mitochondria were monitored using synaptophysin-pHluorin (sypH) and mt-mTagBFP2 in cultured neurons at 20-23DIV. **f** Representative images of sypH peak normalized by $F_{max}$ obtained during NH$_4$Cl (50 mM) incubation. **g–i** Presynaptic release associated with long mitochondria in *Mff* knockdown axons is significantly reduced during 20APs (10 Hz) compared to non-knockdown neurons. Quantification of SypH peak normalize values ($\Delta F/(F_{max} - F_0)$) show significantly reduced neurotransmitter vesicle exocytosis during stimulation of presynaptic release with 20AP at 10 Hz, although total synaptic vesicle pool size ($F_{max}$) is not significantly altered. All images were recorded using the same capturing condition. All graphs are represented with mean ± sem. $n_{control}$ = 16 dishes, 47 boutons; $n_{MFF\ shRNA}$ = 24 dishes, 33 boutons. $p$ = 0.0037, Unpaired *t*-test. Scale bars represent the following lengths: 5 μm

## Discussion

In the present study, we uncovered a novel mechanism by which neurons regulate presynaptic release and axonal development. Our results demonstrate that loss of MFF activity increased mitochondria length by reducing fission for both mitochondria entering the axon as well as along the axonal shaft. Strikingly, mitochondrial elongation did not affect mitochondrial trafficking along the axon, presynaptic positioning, membrane potential, or their ability to produce ATP; however, it dramatically increased their Ca$^{2+}$ buffering capacity. This size-dependent increase in

mitochondrial Ca$^{2+}$ uptake lessened presynaptic cytoplasmic Ca$^{2+}$ levels causing reduced neurotransmitter release upon evoked activity and resulted in decreased terminal axon branching in vivo (Supplementary Fig. 8).

These results suggest a two-step model for MFF-dependent regulation of axonal mitochondrial size where MFF activity is required both at the time of axonal entry as well as for maintenance along the length of the axon. Our data reveals that in control cortical axons, mitochondria enter with an average size of ~1 μm, and maintain this short and strikingly uniform length along the axon by coupling almost every fusion event to a fission event. These results represent the first evidence that axonal mitochondrial size is regulated before and following entry into the axon, and raises intriguing possibilities regarding the underlying mechanism(s). Two non-mutually exclusive scenarios exist: (1) somatic mitochondria are "tagged" as axonal (vs. dendritic) mitochondria; and/or (2) mitochondria entering the axon are "filtered" based on their size. The most compelling evidence supporting the first scenario may be that in hippocampal and cortical neurons axonal and dendritic mitochondria interact with distinct motor adaptor proteins TRAK1 or TRAK2, respectively[48,49]. However, the mechanism and site of first interaction between TRAK1/2 and mitochondria remain unknown, i.e., do they interact in the cell body to determine specificity or are TRAK1/2 themselves targeted to the axonal or dendritic compartments and engaged locally? Evidence for the second scenario and the existence of a size filter only allowing mitochondria of a short length to enter the axon is currently lacking. However, recent work demonstrating the presence of an actin based filter at the axon initial segment (AIS)[50,51] and evidence of myosin motors opposing kinesin-directed mitochondrial movement[52] could represent such a mitochondria size selection filter.

While not the focus of this manuscript, the question of how MFF-dependent fission is regulated differentially in the axon compared to dendrites remains poorly understood and should be the focus of further investigations. In fact, our finding that MFF is present on the outer membrane of both axonal and dendritic mitochondria (Supplementary Fig. 1) excludes the possibility that the high level of MFF-dependent fission in the axon compared to dendrites is not simply due to compartment-specific localization of MFF on mitochondria OMM. Coupled with the results showing no significant change in dendritic mitochondria length upon Mff knockdown (Supplementary Fig. 2), this implies that MFF recruitment, activation or interaction with Drp1 must be differentially regulated in the axonal and somatodendritic compartments. Cortical PNs could accomplish this via: (1) subcellular targeting of different isoforms of MFF, which can be generated through alternative splicing and have different affinities for Drp1[23], (2) MFF can be post-translationally modified through phosphorylation by kinases such as AMPK and "activated" to increase its ability to recruit Drp1[53], (3) although MiD51 and MiD49 are expressed at extremely low levels in neurons, they could potentially alter the MFF/Drp1 interaction from promoting fission to inhibiting fission[54,55], and finally (4) axonal or dendritic Drp1 itself could be post-translationally modified to increase or reduce its ability to interact with MFF and promote fission in a compartment-specific way[6,56–58]. Future investigations will need to test these models.

Our data illustrates that elongated mitochondria produced upon Mff knockdown possessed a comparable membrane potential and maintained similar levels of matrix ATP and redox state (Fig. 5); in fact, based on their increased size, we might expect that they actually have an increased capacity to produce ATP. A previous report observed a requirement for activity-driven ATP synthesis in synaptic function;[10] however, in this report, mitochondria-dependent ATP production could only be observed upon a prolonged and rather non-physiological stimulation protocol (600APs at 10 Hz) while blocking glycolysis[10]. On the other hand, it is now clear that mitochondrial Ca$^{2+}$ buffering is important for synaptic function in more physiological paradigms of evoked neurotransmission[15,16] and we observed significant increases in presynaptic mitochondrial Ca$^{2+}$ uptake upon Mff-knockdown leading to reduced neurotransmitter release following physiological stimulation regimes (20APs @ 10 Hz). In addition, blocking glycolysis or ATP production mainly affected endocytosis of synaptic vesicles (SVs), but not exocytosis[10]. Layer 2/3 cortical neurons are well known to have less spontaneous activity than other PNs;[59] therefore, our results in Mff-deficient axons is more likely to be a result of abnormal release due to increased mitochondria-dependent Ca$^{2+}$ uptake rather than altered ATP production.

A recent study showed that the Bcl-x$_L$-Drp1 complex regulates SV endocytosis, and MFF may form a complex with these proteins[60]. Consistent with our data, Mff knockdown in hippocampal neurons showed reduced exocytosis of SVs. While the authors of this study normalized the pHluorin signal with $F_0$ value from base line signals, we used the $F_{max}$ value obtained by NH$_4$Cl incubation in order to normalize for any potential changes in total vesicle pool size. Even with this more optimal normalization method, MFF-deficient neurons still observed reduced neurotransmitter release compared to control.

In addition to mitochondria, axonal endoplasmic reticulum (ER) was recently shown to be involved in presynaptic Ca$^{2+}$ clearance upon stimulation of neurotransmitter release[61]. Although inhibition of ER Ca$^{2+}$ uptake can change plasma membrane Ca$^{2+}$ entry through STIM1, a recent 3D-EM study revealed the presence of ER and mitochondria contacts along the axon and specifically at presynaptic sites[62]. Future investigation will need to characterize the relative contribution of axonal ER and mitochondria, and ER-mitochondria coupling for presynaptic Ca$^{2+}$ homeostasis.

The role of neuronal activity in axonal development and branching has been well established over the last 20 years. Early work demonstrated that developing neurons require spontaneous activity and spontaneous neurotransmitter release for axonal development[63–65] while more recent work has revealed the requirement for activity on both the presynaptic and postsynaptic sides[35,36,66–68]. Finally, axonal branches which make presynaptic boutons and are synchronized with postsynaptic neuronal activity are more likely to be stabilized when compared to axons with desynchronized activity[69–71]. Based on these previous results and the observation that layer 2/3 terminal branching is the most affected, it seems likely that the loss of terminal axon branching we observe upon Mff knockdown is a result of the decreased neurotransmitter release we observed along these axons.

Taken together, our data demonstrate for the first time that maintenance of small mitochondrial size in CNS axons through MFF-dependent fission is critical to limit presynaptic Ca$^{2+}$ dynamics, neurotransmitter release, terminal axonal branching and therefore proper development of circuit connectivity.

## Methods

**Mice for in utero electroporation.** All animals were handled according to protocols approved by the Institutional Animal Care and Use Committee (IACUC) at Columbia University. Time-pregnant CD1 females were purchased from Charles Rivers. Timed-pregnant hybrid F1 females were obtained by mating inbred 129/SvJ females (Charles Rivers), and C57Bl/6J males (Charles Rivers) in house. At the time of in utero electroporation (Embryonic Day 15.5), littermates were randomly assigned to experimental groups without regard to their sex.

**Mice for ex utero electroporation and primary culture.** All animals were handled according to protocols approved by the Institutional Animal Care and Use

Committee (IACUC) at Columbia University. Time-pregnant CD1 females were purchased from Charles Rivers. At the time of ex utero electroporation (E15.5), littermates were randomly assigned to experimental groups without regard to their sex.

**HEK 293T cells**. Human Embryonic Kidney (HEK) 293T cells were purchased from ATCC.

**Plasmids**. pCAG mt-ATEAM was made by placing the DNA encoding mt-ATEAM (gift from Hiromi Imamura) 3' to the CAG promoter via PCR and Infusion cloning (Clonetech). pCAG mt-Grx1-roGFP2 was made by placing the DNA encoding mt-Grx1-roGFP2 (Addgene plasmid #64977) 3' to the CAG promoter as above. pCAG mScarlet was made by placing the DNA encoding mScarlet (Addgene plasmid #85042) 3' to the CAG promoter as above. pCAG vGLUT1-HA was created by replacing the mCherry in pCAG vGLUT1-mCherry with an HA tag via restriction digest and PCR. pCAG mTAGBFP2 was created by removing the mitochondrial targeting sequence in pCAG mt-mTAGBFP2 via restriction digest and re-ligation. pCAFNF mTAGBFP2 and mt-YFP were made by PCR of the DNA encoding mTAGBFP2 or mt-YFP and placement 3' to the CAG FNF cassette (Addgene plasmid #13772) via PCR and Infusion cloning (Clonetech). pCAG HA-mMff was created via PCR of the DNA encoding mouse *Mff* from a neuronal mouse cDNA library, and sub-cloned 3' to the CAG promoter and a HA tag.

**In utero electroporation**. A mix of endotoxin-free plasmid preparation (2 mg/mL total concentration) and 0.5% Fast Green (Sigma) was injected into one lateral hemisphere of E15.5 embryos using a Picospritzer III (Parker). Electroporation (ECM 830, BTX) was performed with gold paddles to target cortical progenitors in E15.5 embryos by placing the anode (positively charged electrode) on the side of DNA injection and the cathode on the other side of the head. Five pulses of 45 V for 50 ms with 500 ms interval were used for electroporation. Animals were sacrificed 21 days after birth (P21) by terminal perfusion of 4% paraformaldehyde (PFA, Electron Microscopy Sciences) followed by overnight postfixation in 4% PFA. For sparse labeling via in utero electroporation, Flp dependent plasmids (pCAFNF, Addgene) were used along with 640 pg/μL of pCAG Flp-e (Addgene).

**Ex utero cortical electroporation**. A mix of endotoxin-free plasmid preparation (2–5 mg/mL) and 0.5% Fast Green (Sigma) mixture was injected using a Picospritzer III (Parker) into the lateral ventricles of isolated head of E15.5 mouse embryo, and electoporated using an electroporator (ECM 830, BTX) with four pulses of 20 V for 100 ms with a 500 ms interval. Following ex utero electroporation we performed dissociated neuronal culture as described below.

**Primary neuronal culture**. Embryonic mouse cortices (E15.5) were dissected in Hank's Balanced Salt Solution (HBSS) supplemented with HEPES (10 mM, pH 7.4), and incubated in HBSS containing papain (Worthington; 14 U/mL) and DNase I (100 μg/mL) for 20 min at 37 °C. Then, samples were washed with HBSS, and dissociated by pipetting. Cell suspension was plated on poly-D-lysine (1 mg/mL, Sigma)-coated glass bottom dishes (MatTek) or coverslips (BD bioscience) in Neurobasal media (Invitrogen) containing B27 (1 ×), Glutamax (1 ×), FBS (2.5%), and penicillin/streptomycin (0.5 ×, all supplements were from Invitrogen). After 5 to 7 days, media was changed with supplemented Neurobasal media without FBS.

**Immunocytochemistry**. Primary culture: Cells were fixed for 10 min at room temperature in 4% (w/v) paraformaldehyde (PFA, EMS) in PBS (Sigma), then incubated for 30 min in 0.1% Triton X-100 (Sigma), 1% BSA (Sigma), 5% Normal Goat Serum (Invitrogen) in PBS to permeabilize and block nonspecific staining, after washing with PBS. Primary and secondary antibodies were diluted in the buffer described above. Primary antibodies were incubated at room temperature for 1 h and secondary antibodies were incubated for 30 min at room temperature. Coverslips were mounted on slides with Fluoromount G (EMS). Primary antibodies used for immunocytochemistry in this study are chicken anti-GFP (5 μg/mL, Aves Lab—recognizes GFP and YFP), mouse anti-HA (1:500, Covance), rabbit anti-RFP (1:1,000, Abcam–recognizes mTagBFP2, DsRED, and tdTomato). All secondary antibodies were Alexa-conjugated (Invitrogen) and used at a 1:2000 dilution. Nuclear DNA was stained using Hoechst 33258 (1:10,000, Pierce)

Endogenous MFF staining: Cells were fixed for 10 min at room temperature in 4% PFA in PBS, followed directly with – 20 °C, 100% methanol at – 20 °C for 6 min. After washing with room temperature PBS, cells were incubated for 30 min in 1% BSA (Sigma), 5% Normal Goat Serum in PBS to block nonspecific staining. Primary and secondary antibodies were diluted in the buffer above. Primary antibodies were incubated at room temperature for 1 h and secondary antibodies were incubated for 30 min at room temperature. Coverslips were mounted on slides with Fluoromount G (EMS). Primary antibodies used for this section are chicken anti-GFP (5 μg/mL, Aves Lab—recognizes GFP and YFP), rabbit anti-MFF (1:200, Protein Tech). All secondary antibodies were Alexa-conjugated (Invitrogen) and used at a 1:2000 dilution.

Brain sections: Post fixed brains were sectioned via vibratome (Leica VT1200) at 100 μm. Floating sections were then incubated for 2 h in 0.4% Triton X-100, 1%

BSA, 5% Normal Goat Serum in PBS to block nonspecific staining. Primary and secondary antibodies were diluted in the buffer described above. Primary and secondary antibodies were incubated at 4 °C overnight. Sections were mounted on slides and coversliped with Aqua PolyMount (Polymount Sicences, Inc). Primary and secondary antibodies are the same as above.

**Imaging**. Fixed samples were imaged on a Nikon Ti-E microscope with an A1 confocal. All equipment and solid state lasers (Coherent, 405, 488, 561, and 647 nm) were controlled via Nikon Elements software. Nikon objectives used include 20 × (0.75NA), 40 × (0.95NA) or 60x oil (1.4NA). Optical sectioning was performed at Nyquist for the longest wavelength. Analysis of mitochondrial length and occupancy were performed in Nikon Elements.

Live imaging: Electroporated cortical neurons were imaged at 7-21DIV with EMCCD camera (Andor, iXon3-897) on an inverted Nikon Ti-E microscope (40x objective NA0.95 with 1.5x digital zoom or 60x objective NA1.4) with Nikon Elements. In all, 488 and 561 nm lasers shuttered by Acousto-Optic Tunable Filters (AOTF) or 395, 470, and 555 nm Spectra X LED lights (Lumencor) were used for the light source, and a custom quad-band excitation/dichroic/emission cube (based off Chroma, 89400) followed by clean up filters (Chroma, ET435/26, ET525/50, ET600/50) were applied for excitation and emission. We used modified normal tyrode solution as a bath solution at 37 °C (Tokai Hit Chamber), which contained (in mM): 145 NaCl, 2.5 KCl, 10 HEPES pH 7.4, 2 $CaCl_2$, 1 $MgCl_2$, 10 glucose.

For ATEAM (gift from Hiromi Imamura) imaging a CFP/YFP cube (Chroma, 59217) followed by clean up filters (Chroma, ET475/20, ET540/21) was used, while for roGFP2 (Addgene) imaging, the custom cube above was used followed by clean up with ET525/50.

For Tetramethylrhodamine (Sigma, TMRM) imaging cells were incubated with 10 nM TMRM for 20 min at 37 °C before imaging started.

For calcium imaging on evoked release, we added APV (50 μM, Tocris) and CNQX (20 μM, Tocris) in bath solution. Evoked releases were triggered by 1 ms current injections with a concentric bipolar electrode (FHC) placed 20 μm away from transfected axons. We applied 20APs at 10 Hz with 30 V using the stimulator (Model 2100, A-M systems) and imaged with 500 ms or 100 ms interval (2 Hz or 10 Hz) during 90 s for mt-GCaMP5G signals and 100 ms interval (10 Hz) for 15 s for VGLUT1-GCaMP5G signals. At the end of acquiring, we added the calcium ionophore ionomycin (5 μM, EMD Millipore) and continued imaging with 1 s interval to obtain $F_{max}$ value. For blocking mitochondrial $Na^+/Ca^{2+}$ exchanger, we added CGP 37157 (10 μM, Tocris) following mt-GCaMP5G imaging with 500 ms interval. After 3 min, we imaged the same area with the identical condition (20APs at 10 Hz).

For syp-pHluorin imaging, 20APs were applied at 10 Hz and neurons were imaged with 100 ms interval (10 Hz) during 60 s, then the bath solution was changed with tyrode solution containing 55 mM $NH_4Cl$ for $F_{max}$ value.

Images were analyzed using a Fiji (Image J) plug-in, Time Series Analyzer (v3.0). Each vGLUT1-GCaMP5G or syp-pHluorin puncta and nearby backgrounds were selected by circular region of interests (ROIs) and intensities were measured by plug-in. For mt-GCaMP5G measurement, single synapse-associated mitochondria were pooled for comparing uptake of presynaptic $Ca^{2+}$ from a single presynaptic bouton. Full-length mitochondria were marked by a freehand selection tool and total intensities were measured. After intensities were corrected for background subtraction, $\Delta F$ values were calculated from $(F - F_0)$. $F_0$ values were defined by averaging 10 frames before stimulation, and $F_{max}$ values were determined by averaging 10 frames of maximum plateau values following ionomycin or $NH_4Cl$ application, then, used for normalization. Long mitochondria in *Mff* knockdown axons for $Ca^{2+}$ and pHluorin imaging were selected by the length over 3 μm. Diffusion of long mitochondrial $Ca^{2+}$ from a single presynapse is also analyzed by Fiji. Line ROIs (1 μm, width 3) were placed on a presynapse-overlapped site and every 2 μm distance from the site. Then, Plot Z-axis Profile function was performed for obtaining intensities from each line, and time to peak was defined from the plot.

**Cell line transfection and western blot**. HEK cells were transfected via jetPrime (Polyplus) by following the manufacture's instructions. Cells were harvested in cold DPBS and lysed in ice-cold lysis buffer containing 25 mM Tris (pH 7.5), 2 mM $MgCl_2$, 600 mM NaCl, 2 mM EDTA, 0.5% NP-40, 1X protease and phosphatase cocktail inhibitors (Sigma) and Benzonase (0.25U/μL of lysis buffer; Novagen). Aliquots of the proteins were separated by SDS-PAGE and then transferred to a polyvinylidene difluoride (PVDF) membrane (Amersham). After transfer, the membrane was washed 3X in Tris Buffer Saline (10 mM Tris-HCl pH 7.4, 150 mM NaCl) with 0.1% of Tween 20 (T-TBS), blocked for 1 h at room temperature in Odyssey Blocking Buffer (TBS, LI-COR), followed by 4 °C overnight incubation with the appropriate primary antibody in the above buffer. The following day, the membrane was washed 3X in T-TBS, incubated at room temperature for 1 h with IRDye secondary antibodies (LI-COR) at 1:10,000 dilution in Odyssey Blocking Buffer (TBS), followed by 3X T-TBS washes. Visualization was performed by quantitative fluorescence using an Odyssey CLx imager (LI-COR). Signal intensity was quantified using Image Studio software (LI-COR). Primary antibodies used for western-blotting are mouse anti-HA (1:2000, Covance), rabbit anti-ERp72 (1:1000, Cell Signaling Technologies).

**Quantification and statistical analysis**. All statistical analysis and graphs were performed/created in Graphpad's Prism 6. Statistical tests, $p$-values, and ($n$) numbers are presented in the figure legends. Gaussian distribution was tested using D'Agostino & Pearson's omnibus normality test. We applied non-parametric tests when data from groups tested deviated significantly from normality. All analysis were performed on raw imaging data without any adjustments. Images in figures have been adjusted for brightness and contrast (identical for control and experimental conditions in groups compared).

## Data availability

Supporting data of this study are available from the corresponding author on reasonable request.

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

## Acknowledgements

We would like to thank members of the F.P. and R.S. labs for helpful suggestions, technical advice, and critical reading of the manuscript. Funding sources include NIH-R01NS089456 (F.P.), NIH-K99NS091526 (T.L.L.), the KIST Institutional Program 2E27850 (S.K.), Human Frontiers Science Program Long-term Fellowship (S.K.), and an Award from the Fondation Roger De Spoelberch (F.P.), NIH-R35CA220538. (R.S.), NIH-P01CA120964 (R.S.) and NIH-P30CA014195 (R.S.).

## Author contributions

T.L.L., R.S., and F.P. contributed to the conceptualization of this manuscript. T.L.L., S.K., and F.P. contributed to the methodology of this manuscript. T.L.L., S.K., and A.L. contributed to the investigation presented in this manuscript. T.L.L., S.K., and F.P. participated in the writing of this manuscript. T.L.L. and F.P. secured funding for the work presented in this manuscript. R.S. provided resources which enabled the work presented in this manuscript.

## Additional information

**Competing interests:** The authors declare no competing interests.

