## [Peer Review File · Nature Communications]

Reviewers' comments:

Reviewer #1 (Remarks to the Author):

Summary:

In the current manuscript, Tommy et al. investigated the functional importance of mitochondrial fission for axonal development. The authors found that mitochondrial morphology is different in the axonal and dendritic compartments of cortical neurons and axonal mitochondrial size is dependent on 'Drp1 receptor' Mitochondrial fission factor (MFF). This study suggests that downregulation of Mff decreases fission and hence increases axonal mitochondria size/connectivity accompanied by increased $[Ca^{2+}]_m$ uptake at the presynaptic site to enhance local intracellular calcium buffering and thereby reduces neurotransmitter release and axonal branching. This study demonstrates a novel mechanism governing axon branching and neurotransmitter release via Mff mediated regulation of mitochondrial size.

Overall, excellent work by an outstanding lab.

Concerns:

1. The authors suggest that MFF-dependent mitochondrial fission in axons does not alter oxidative capacity and ATP production based on the A-Team assay. Previous reports suggest that mitochondrial shape has an impact on mitochondrial bioenergetics. For example, recently Zhou et al. 2017 showed that Mff knockout in cardiomyocytes have distinct bioenergetics effect on the mitochondria and Toyoma et al., 2016 showed regulation of Mff activity by energy sensing AMPK, further suggest the direct crosstalk between the Mff activity and energy homeostasis of the cells. More data is needed to support the notion that there is no change in bioenergetics and that this is merely a buffering effect of mitochondria. While I acknowledge the difficulty in monitoring OCR in axon vs. dendrite populations any additional experiments to monitor OxPhos capacity or ATP generation would aid this interpretation.
2. Along these same lines, in Figure 6, axonal mitochondrial elongation upon Mff-knockdown showed increased mitochondrial calcium uptake. Mitochondrial calcium uptake and dynamics is known to couple with TCA cycle and increase oxidative capacity, but there is no change in bioenergetics upon Mff knock down. Please discuss and explain.
3. One key experiment would be to examine this same phenomenon in Mcu cKO mice/neurons, which would prove that the mitochondrial uptake/buffering is the key physiological difference downstream of the shape changes.
4. The manuscript lacks any data to evaluate the expression or function of Drp1 levels in axonal and dendritic mitochondria. It is well established that Drp1 interacts with Mff to mediate fission. It'd be nice to see if Drp1 knockdown/knockout has a reciprocal effect on the axonal/dendritic mitochondrial shape.
5. Can the authors explain how they determined individual mitochondrial length in the dendrite where the mitochondria appear as a network? Should this read mitochondrial connectivity or uninterrupted network length? Were other morphological parameters examined?
6. How was the mtGCaMP5G signal corrected? Qualitatively, it looks as if there is an equal change in signal in the Mff-shRNA group as control along the axon and particularly at the presynaptic region (Figure 6a-b). Since this is not a ratiometric reporter it's important to perhaps also show the data not corrected to baseline, especially since the signal goes from 0 to very robust at 2 s in the Mff -shRNA group whereas in the control group there is an obvious signal at baseline. Is the basal mitochondrial

content different with MFF knockdown? The use of a ratiometric reporter may help examine this possibility.

Minor

1. The figure legends need more information to aid the reader. Please define constructs, etc. and add more details so the figures stand-alone.

2. Not sure why the experiment with the NCLX inhibitor was performed unless the amplitude changed. i.e. was an increase in NCLX-mediated efflux somewhat masking the increase in MCU-mediated uptake?

Reviewer #2 (Remarks to the Author):

The manuscript by Lewis et al describes the role of the Drp1 receptor MFF in controlling mitochondria size in axons of cortical pyramidal neurons. The authors show that MFF depletion, both in vitro and in vivo, increases presynaptic mitochondrial size and that this correlates with augmented capacity for calcium uptake during neurotransmission reducing presynaptic calcium levels, synaptic vesicle release and axonal branching. The authors conclude that fission-dependent regulation of mitochondrial size is a novel mechanism to control axonal branching and synaptic vesicle release.

Mitochondria play an important role in neuronal physiology both at pre- and postsynaptic sites. Previous studies, also from the authors' lab, have shown that ATP generation and calcium buffering by presynaptic mitochondria controls efficiency of synaptic vesicle release. In addition, striking differences are observed in mitochondrial morphology in axons (small, punctate) and dendrites (long and tubular) but the physiological consequence remains to be determined. Hence, the finding that one of the mitochondrial fission factors, MFF, appears to control axonal mitochondria morphology and presynaptic function is clearly of interest. In figures 1 and 2 the authors convincingly show that in vitro and in vivo depletion of MFF leads to elongation of axonal mitochondria via a reduction in mitochondrial fission (figure 3).

However, the analysis of the functional implications of MFF depletion is sometimes anecdotal (figure 4) and suffers from over interpretation of the data (figures 6 and 7). Moreover, the manuscript does not provide mechanistic insight in how MFF-dependent fission is specific for axonal mitochondria as MFF is also present on dendritic mitochondria.

Major concerns

(A) Concerning experimental design testing causality between effect on presynaptic calcium and synaptic vesicle release upon MFF depletion.

In figure 6 the authors show that calcium uptake is increased in elongated mitochondria. In figure 7 the authors try to show causality between increased calcium uptake and decreased presynaptic calcium levels and synaptic vesicle release by comparing control versus MFF depleted neurons. This is not the correct analysis. To test if the presence of an elongated mitochondria affects synaptic vesicle release at a single synapse (as the authors claim), the authors should compare synapses with and without elongated mitochondria in MFF depleted neurons and not the average response of all synapses between control and MFF depleted neurons. This is important as the current data in figure 7 tests the OVERALL effect of MFF depletion, which could be the result of decreased neuronal health upon MFF depletion. But it does not allow the conclusion that "presynaptic release at sites of elongated

mitochondria is impaired in Mff knockdown neurons". As stated above, to make this statement the authors should compare synapses plus and minus elongated mitochondria in Mff depleted neurons (a similar analysis as in their previous paper by Kwon et al., 2016).

(B) Selection bias due to low number of observations per neuron

In addition to my comments in (A). In figure 7 the authors measure the effect of MFF depletion on synaptic vesicle release in 33 boutons from 24 dishes. This amounts to 1-2 boutons per dish. What is the reason for this extremely low number of boutons analyzed per dish? Even with only 1 labeled neuron per dish the authors should be able to analyse >100 synapses per dish. How did the authors select the 1-2 boutons out of the >100 synapses? How did the authors prevent selection bias etc etc.

(C) Over interpretation of the lack of effect of FIS1

The authors conclude from Figure 2 that MFF but not FIS1 is required for maintenance of small mitochondrial size in axons. However, the characterization of shRNA efficiency in heterologous cells does not allow such a conclusion. In their HEK cell experiment, the FIS1 shRNAs appear to be much less efficient than MFF shRNAs (Fig. S1). More importantly, the effect on endogenous FIS1 is not evaluated. Without a more robust analysis of endogenous FIS1 depletion the conclusion that FIS1 does not affect mitochondrial size in axons is an over interpretation.

(D) data presented as ratios without showing primary data of Fmax values

In figure 7 normalized data is shown of [Ca]_{cyto} and SyPhy fluorescence without the corresponding Fmax data. Does MFF depletion have an effect on Fmax upon 5 M ionomycin treatment (Figure 7a-d) or on SyPhy Fmax upon NH₄Cl application (Figure 7e-g)? Without this information it is difficult to assess potential confounding factors such as effects on total vesicle numbers as this will affect VG1GC and SyPhy targeting (both present on synaptic vesicles). The authors should show these primary data in figure 7.

(E) The author's working model is not supported by the data

As I commented before (comment A), the authors did not test whether the presence of an elongated mitochondria affects synaptic vesicle release at that site. This working model is therefore not supported by the data.

Minor

(A) concerning validation of mitochondrial probes. The authors do not show the effect of MFF depletion using endogenous markers of mitochondria but instead use a vast array of mitochondrial probes in many different combinations. To rule out any effects of these probes staining with endogenous mitochondrial markers showing the effect of MFF depletion would strengthen the authors' conclusion.

Reviewer #3 (Remarks to the Author):

This paper provides evidence that MFF mediates the size of axonal mitochondria, and that control of mitochondrial size has implications for Ca²⁺ buffering in nerve terminals and thus neurotransmitter release. The authors propose that MFF controls the size and number of mitochondria exiting the soma to traffic along the axon, and that this affects terminal axon branching in vivo.

Overall, this work should be of broad appeal to neuroscientists, and provide new and important advances to our understanding of synaptic biology. However, some additional data is required to support the claims made by the authors.

Major

1) This paper relies entirely on the use of Mff shRNA. Whilst strong evidence has been provided that suggests that these shRNA are effective, no quantification of knockdown efficiency is presented. 1 Western blot is provided (with a single n of each shRNA), and representative immunolabelled neurons (for Mff but not for Fis1). Authors should provide quantification of knockdown efficiency for both Mff and Fis1 from immunolabelled neurons (i.e. electroporated vs non-electroporated from same field of view) to indicate efficiency of knockdown in their model system.

In main methods and results text, authors specify that 'a mixture of shRNA' is used for knockdown. In supplementary fig 1, 3x (triple) shRNA have been used in HEK cells/ Westerns, and 2x shRNA (539+665) have been used in neurons. Authors should clarify in main text which shRNA have been used for all experiments, and should quantify the efficiency of knockdown from that system (i.e. if 539+665 have been used throughout, 539+665 should be quantified).

2) Related to above, Fis1 knockdown looks less efficient from the Western blot, and no evidence provided of knockdown efficiency in neurons. Therefore, relative lack of effect of Fis1 knockdown on mitochondrial length in Fig 2 may be due to less efficient knockdown.

3) More details are required regarding the assay in Fig 3. It is clear how this would be used to measure fusion; how does it measure fission?

4) In Fig 7, the peak height of pHLuorin fluorescence represents an equilibrium between exocytosis and compensatory endocytosis occurring during stimulation. Thus, the difference between the two peaks may be driven by reduction in exocytosis, or by an increased rate of endocytosis during the stimulus train.

To confirm that this is due to a reduction in evoked exocytosis, this assay should be performed in the presence of a vATPase inhibitor (eg bafilomycin) to block vesicle reacidification and allow quantification of exocytosis specifically.

5) In the model diagram, the authors propose that in terminals lacking mitochondria, there is an increased Ca^{2+} build-up, and increased neurotransmitter release (S7 C and D, left). In terminals with normal mitochondria, there is reduced neurotransmitter release (S7C right), and in terminals with large mitochondria, there is a strong reduction in neurotransmitter release (S7D right). This is an elegant proposal, but evidence for this is not presented in this paper. Instead, authors have demonstrated that in terminals with long mitochondria there is a reduction in peak height of fluorescence during stimulation, compared to terminals with normal mitochondria (however this requires clarification – see minor comments). Moreover, several studies have shown that neurotransmitter release is similar in synapses with and without mitochondria (eg Pathak et al JBC 2015).

Authors could present evidence for their hypothesis, by comparing not only terminals with long/short mitochondria (control vs shMFF), but also comparing with/without mitochondria from within the same axon – this data will be available through the exocytosis assay for Fig 7.

Minor

Used throughout: vesicle release. Synaptic vesicles aren't 'released'. Should be neurotransmitter release, or synaptic vesicle exocytosis.

Please clarify what is meant 'high density' cultures (line 110)

Please clarify what is meant by 'control' in Fig 7F: is control non-knockdown neurons with small mitochondria? Or terminals without mitochondria?

Please clarify throughout how many cultures/animals have been used. This is sometimes presented (eg. Fig 2 line 687) but has not been uniformly presented for all figures.

Response to reviewers - NCOMMS-18-06119

Reviewer #1 (Remarks to the Author):

Summary:

In the current manuscript, Tommy et al. investigated the functional importance of mitochondrial fission for axonal development. The authors found that mitochondrial morphology is different in the axonal and dendritic compartments of cortical neurons and axonal mitochondrial size is dependent on 'Drp1 receptor' Mitochondrial fission factor (MFF). This study suggests that downregulation of Mff decreases fission and hence increases axonal mitochondria size/connectivity accompanied by increased $[Ca^{2+}]_m$ uptake at the presynaptic site to enhance local intracellular calcium buffering and thereby reduces neurotransmitter release and axonal branching. This study demonstrates a novel mechanism governing axon branching and neurotransmitter release via Mff mediated regulation of mitochondrial size. Overall, excellent work by an outstanding lab.

We appreciate and thank the reviewer for their remarks and enthusiasm. We have done our best to address each of his/her suggestions below.

Concerns:

1. The authors suggest that MFF-dependent mitochondrial fission in axons does not alter oxidative capacity and ATP production based on the A-Team assay. Previous reports suggest that mitochondrial shape has an impact on mitochondrial bioenergetics. For example, recently Zhou et al. 2017 showed that Mff knockout in cardiomyocytes have distinct bioenergetics effect on the mitochondria and Toyoma et al., 2016 showed regulation of Mff activity by energy sensing AMPK, further suggest the direct crosstalk between the Mff activity and energy homeostasis of the cells. More data is needed to support the notion that there is no change in bioenergetics and that this is merely a buffering effect of mitochondria. While I acknowledge the difficulty in monitoring OCR in axon vs. dendrite populations any additional experiments to monitor OxPhos capacity or ATP generation would aid this interpretation.

We agree that monitoring OCR in the dendrite and axonal populations is challenging, and this is further complicated by the fact that these mature cultures contain a mixture of cell types (both different neuronal subtypes and astrocytes) in order to produce mature, synaptically active and healthy neurons. This is the major reason we have relied on genetically encoded probes to label the specific population of cortical neurons born at E15.5. However, we also agree that more could be done; therefore, we have included **new data in Figure S5** quantifying mitochondrial membrane potential in individual axonal mitochondria using TMRM fluorescence intensities (normalized to genetically encoded fluorescent blue protein targeted to mitochondrial matrix ; mito-mTagBFP2). We measured baseline normalized TMRM fluorescence in control and MFF shRNA expressing cortical pyramidal neurons and then treated with Antimycin A (complex III inhibitor) to visualize the fraction of mitochondrial membrane potential resulting from oxidative phosphorylation (OxPhos). We find that Antimycin A treatment leads to approximately 30% reduction in membrane potential in both control and MFF knockdown conditions (**Fig S5e-f**). This new data again argues that these axonal mitochondria in control and shMFF expressing cortical PNs are performing similar levels of Oxphos. In each of our experiments, we have attempted to follow guidelines set forth by the field ¹.

One distinct difference between neurons and other cell types is that AMPK is mainly used as a stress response pathway and activated by a different upstream kinase (CAMKK2 in neurons vs.

LKB1 in other cell types)^{2,3}. This may partially explain the differences we observe compared with those seen in other cell types.

2. Along these same lines, in Figure 6, axonal mitochondrial elongation upon Mff-knockdown showed increased mitochondrial calcium uptake. Mitochondrial calcium uptake and dynamics is known to couple with TCA cycle and increase oxidative capacity, but there is no change in bioenergetics upon Mff knock down. Please discuss and explain.

We agree on this point. Our stimulation condition for Ca²⁺ uptake is 20 action potentials (AP) delivered at 10Hz. Although previous studies showed that much stronger, non-physiological, stimulation (600APs) induces activity-dependent ATP generation at presynaptic sites⁴, bioenergetics upon short-term stimulation in neurons is still not well characterized and the effect might be more limited. In addition, recent studies suggest that glycolysis is involved in activity-dependent presynaptic ATP generation^{5,6}. Therefore, activity-dependent mitochondrial Ca²⁺ uptake using a physiological range of stimulation may not have a detectable effect on overall ATP generation by presynaptic mitochondria. We added a discussion point to this effect.

3. One key experiment would be to examine this same phenomenon in Mcu cKO mice/neurons, which would prove that the mitochondrial uptake/buffering is the key physiological difference downstream of the shape changes.

We appreciate this suggestion. We are in the process of acquiring and crossing the MCU floxed mouse line (which will take up several more months), but in the meantime, we determined the presynaptic cytoplasmic Ca²⁺ dynamics evoked during AP-mediated stimulation of neurotransmitter release in MFF knockdown neurons in the presence of an MCU inhibitor (Ru360). As predicted by the idea that the ability of mitochondria to buffer cytoplasmic Ca²⁺ through MCU channel impacts presynaptic Ca²⁺ accumulation during trains of AP⁷, MCU application significantly elevated presynaptic Ca²⁺ levels at presynaptic boutons associated with elongated mitochondria compared to control. This data has been added to new **Fig. S7d-e**.

4. The manuscript lacks any data to evaluate the expression or function of Drp1 levels in axonal and dendritic mitochondria. It is well established that Drp1 interacts with Mff to mediate fission. It'd be nice to see if Drp1 knockdown/knockout has a reciprocal effect on the axonal/dendritic mitochondrial shape.

We agree that Drp1 is an important mediator of Mff dependent fission, however work from other labs has shown that Drp1 disruption effects mitochondria drastically in the cell body and even prevents mitochondria from entering the axon⁸. We also feel this is outside the scope of the paper as we are using Mff knockdown as a tool to study the role of mitochondrial size in the axon of cortical pyramidal neurons and not to determine the mechanism underlying the differential regulation of mitochondrial size in these two compartments. The mechanism is of course very important but also very complex and will take much more work to parse out.

5. Can the authors explain how they determined individual mitochondrial length in the dendrite where the mitochondria appear as a network? Should this read mitochondrial connectivity or uninterrupted network length? Were other morphological parameters examined?

We used high magnification confocal imaging to image the mitochondria in the x,y and z dimensions. To display these images, we need to use maximum projections but for quantitative analysis we used the z-dimension to distinguish areas of overlap between mitochondria. Length

and fraction of axonal or dendritic volume occupied by mitochondria were the two factors measured in vitro and in vivo (see for example quantification in **Fig. 1g-h**).

6. How was the mtGCaMP5G signal corrected? Qualitatively, it looks as if there is an equal change in signal in the Mff-shRNA group as control along the axon and particularly at the presynaptic region (Figure 6a-b). Since this is not a ratiometric reporter it's important to perhaps also show the data not corrected to baseline, especially since the signal goes from 0 to very robust at 2 s in the Mff-shRNA group whereas in the control group there is an obvious signal at baseline. Is the basal mitochondrial content different with MFF knockdown? The use of a ratiometric reporter may help examine this possibility.

mt-GCaMP5G signals were corrected by baseline subtraction. Because we wanted to estimate the total amount ('quantal content') of Ca^{2+} imported into the mitochondrial matrix of control and elongated mitochondria upon Mff knockdown, i.e. take into account the effect of increased mitochondrial matrix volume, we measured integrated intensity from full-length mitochondria. In addition, when we compared baseline signals between small mitochondria and elongated ones, elongated ones have higher total Ca^{2+} level (new **Fig. S7a**), largely reflecting the increase in size since when we normalized F0 values by mitochondrial size, we find no difference between control and Mff-knockdown F0 values (new **Fig. S7b**).

Minor

1. The figure legends need more information to aid the reader. Please define constructs, etc. and add more details so the figures stand-alone.

We have added additional details and attempted to standardize the figure legends.

2. Not sure why the experiment with the NCLX inhibitor was performed unless the amplitude changed. i.e. was an increase in NCLX-mediated efflux somewhat masking the increase in MCU-mediated uptake?

As this reviewer mentioned, elongated mitochondria in MFF knockdown axons have faster decay (new panel in **Fig. 6e**) likely because of their larger size. We added NCLX inhibitor to isolate the effect of mitochondrial Ca^{2+} uptake since $\text{Na}^+/\text{Ca}^{2+}$ exchanger-dependent extrusion of Ca^{2+} from mitochondria co-occurs with MCU-dependent Ca^{2+} influx, we wanted to make sure that the increased 'quantal content' for Ca^{2+} import in elongated mitochondria was not obscured by changes in $\text{Na}^+/\text{Ca}^{2+}$ exchanger activity. We added more details in the text.

Reviewer #2 (Remarks to the Author):

The manuscript by Lewis et al describes the role of the Drp1 receptor MFF in controlling mitochondria size in axons of cortical pyramidal neurons. The authors show that MFF depletion, both in vitro and in vivo, increases presynaptic mitochondrial size and that this correlates with augmented capacity for calcium uptake during neurotransmission reducing presynaptic calcium levels, synaptic vesicle release and axonal branching. The authors conclude that fission-dependent regulation of mitochondrial size is a novel mechanism to control axonal branching and synaptic vesicle release.

Mitochondria play an important role in neuronal physiology both at pre- and postsynaptic sites. Previous studies, also from the authors' lab, have shown that ATP generation and calcium buffering by presynaptic mitochondria controls efficiency of synaptic vesicle release. In addition, striking differences are observed in mitochondrial morphology in axons (small, punctate) and dendrites (long and tubular) but the physiological consequence remains to be determined. Hence, the finding that one of the mitochondrial fission factors, MFF, appears to control axonal mitochondria morphology and presynaptic function is clearly of interest. In figures 1 and 2 the authors convincingly show that in vitro and in vivo depletion of MFF leads to elongation of axonal mitochondria via a reduction in mitochondrial fission (figure 3).

However, the analysis of the functional implications of MFF depletion is sometimes anecdotal (figure 4) and suffers from over interpretation of the data (figures 6 and 7). Moreover, the manuscript does not provide mechanistic insight in how MFF-dependent fission is specific for axonal mitochondria as MFF is also present on dendritic mitochondria.

We appreciate and thank the reviewer for his/her remarks and enthusiasm. We have done our best to address each of their suggestions below.

Major concerns

(A) Concerning experimental design testing causality between effect on presynaptic calcium and synaptic vesicle release upon MFF depletion.

In figure 6 the authors show that calcium uptake is increased in elongated mitochondria. In figure 7 the authors try to show causality between increased calcium uptake and decreased presynaptic calcium levels and synaptic vesicle release by comparing control versus MFF depleted neurons. This is not the correct analysis. To test if the presence of an elongated mitochondria affects synaptic vesicle release at a single synapse (as the authors claim), the authors should compare synapses with and without elongated mitochondria in MFF depleted neurons and not the average response of all synapses between control and MFF depleted neurons. This is important as the current data in figure 7 tests the OVERALL effect of MFF depletion, which could be the result of decreased neuronal health upon MFF depletion. But it does not allow the conclusion that "presynaptic release at sites of elongated mitochondria is impaired in Mff knockdown neurons".

As stated above, to make this statement the authors should compare synapses plus and minus elongated mitochondria in Mff depleted neurons (a similar analysis as in their previous paper by Kwon et al., 2016).

We agree with this reviewer's concern. As suggested, we have added new analysis comparing presynaptic Ca^{2+} dynamics at presynaptic boutons with and without elongated mitochondria in MFF-deficient axons (**Fig. S7b**). This new analysis demonstrates that decreased presynaptic Ca^{2+} accumulation during evoked release is limited to mitochondria-associated boutons rather than overall effect of MFF depletion.

(B) Selection bias due to low number of observations per neuron. In addition to my comments in (A). In figure 7 the authors measure the effect of MFF depletion on synaptic vesicle release in 33 boutons from 24 dishes. This amounts to 1-2 boutons per dish. What is the reason for this extremely low number of boutons analyzed per dish? Even with only 1 labeled neuron per dish the authors should be able to analyze >100 synapses per dish. How did the authors select the 1-2 boutons out of the >100 synapses? How did the authors prevent selection bias etc etc.

We apologize for the lack of clarity on why these numbers seems low 'per dish'. First of all, as the reviewer might appreciate these experiments are quite technically challenging. The signals measured have low fluorescence, occurs in the 10-100 msec time range, and requires each neuron be approached by an individual bipolar electrode in order to induce a fixed number of action potentials to evoke presynaptic release along its axon. In particular, from an imaging standpoint, we used long-term (~21DIV) dissociated cortical cultures in order to examine presynaptic boutons at 'mature' synapses, and for this purpose, we performed 'classic' co-cultures of cortical neurons with astrocytes. This adds a layer of cells between glass and the axonal boutons, which interferes with the proper focus of every presynaptic bouton in the field of view, therefore reducing the total number of individual presynaptic boutons that can be imaged at high resolution with our CCD camera. In addition, we performed labeling using *ex utero* electroporation and this method electroporates a limited number of pyramidal neurons (<5%) but provides optically-isolated axons. Finally, and most importantly, for normalization of the genetically encoded sensors, at the end of each imaging session for each set of presynaptic boutons along the imaged axon segment, we applied the ionophore ionomycin (for VGlut1-GCaMP5G) or NH₄Cl (for syp-pHluorin), which restricts each dish to one field of view per coverslip. Using this experimental approach, finding presynaptic boutons overlapped with elongated mitochondria was very challenging in the Mff knockdown experiments. Therefore, we obtain a limited number of presynaptic boutons per coverslip but overall, we performed these experiments dozens of times in at least 3 independent cultures in order to accumulate a robust number of individual presynaptic boutons for each experimental group.

(C) Over interpretation of the lack of effect of FIS1

The authors conclude from Figure 2 that MFF but not FIS1 is required for maintenance of small mitochondrial size in axons. However, the characterization of shRNA efficiency in heterologous cells does not allow such a conclusion. In their HEK cell experiment, the FIS1 shRNAs appear to be much less efficient than MFF shRNAs (Fig. S1). More importantly, the effect on endogenous FIS1 is not evaluated. Without a more robust analysis of endogenous FIS1 depletion the conclusion that FIS1 does not affect mitochondrial size in axons is an over interpretation.

As the results were negative, we realized that the Fis1 data was not necessary to the major objective and findings presented in the paper, therefore we have removed this data from the revised version and claims associated with it from the manuscript.

(D) data presented as ratios without showing primary data of Fmax values

In figure 7 normalized data is shown of [Ca]_{cyto} and SyPhy fluorescence without the corresponding F_{max} data. Does MFF depletion have an effect on F_{max} upon 5 M ionomycin treatment (Figure 7a-d) or on SyPhy F_{max} upon NH₄Cl application (Figure 7e-g)? Without this information it is difficult to assess potential confounding factors such as effects on total vesicle numbers as this will affect VG1GC and SyPhy targeting (both present on synaptic vesicles). The authors should show these primary data in figure 7.

We also agree that this information is important for better interpretation. We added F_{max} values for VGlut1-GCaMP5G and syn-pHluorin in **Figure 7e and i**.

(E) The author's working model is not supported by the data

As I commented before (comment A), the authors did not test whether the presence of an elongated mitochondria affects synaptic vesicle release at that site. This working model is therefore not supported by the data.

We have added new data to demonstrate that along the same axon, elongated mitochondria do affect presynaptic calcium and neurotransmitter release (**Fig. S7**). We believe this new data strongly supports our working model and main conclusion that increased in presynaptic mitochondrial size upon inhibition of Mff-dependent fission leads to increased calcium uptake capacity, reduced presynaptic cytoplasmic calcium accumulation during physiological trains of AP (20AP at 10Hz) and reduced induced neurotransmitter vesicle release.

Minor

(A) concerning validation of mitochondrial probes. The authors do not show the effect of MFF depletion using endogenous markers of mitochondria but instead use a vast array of mitochondrial probes in many different combinations. To rule out any effects of these probes staining with endogenous mitochondrial markers showing the effect of MFF depletion would strengthen the authors' conclusion.

We have included new data to show validation of the TMRM-dependent mitochondrial membrane potential measurement (**Fig S5e**) and mt-roGFP2 probe for measuring mitochondrial redox potential (**Fig S5g-h**). We would also argue that TMRM is a field standard, while each of the probes have been used by multiple different labs in neurons either in vitro or in vivo^{9 10 11 12}. Much of the reason we have relied on genetically encoded probes stems from the fact that these mature cultures are a mixture of multiple cells types and therefore it is difficult to distinguish individual axonal/presynaptic mitochondria in a cell-type specific way (only progenitors of layer 2/3 pyramidal neurons electroporated either by in utero or ex utero electroporation at E15).

Reviewer #3 (Remarks to the Author):

This paper provides evidence that MFF mediates the size of axonal mitochondria, and that control of mitochondrial size has implications for Ca²⁺ buffering in nerve terminals and thus neurotransmitter release. The authors propose that MFF controls the size and number of mitochondria exiting the soma to traffic along the axon, and that this affects terminal axon branching in vivo.

Overall, this work should be of broad appeal to neuroscientists, and provide new and important advances to our understanding of synaptic biology. However, some additional data is required to support the claims made by the authors.

We appreciate and thank the reviewer for their remarks and enthusiasm. We have done our best to address each of their suggestions below.

Major

1) This paper relies entirely on the use of Mff shRNA. Whilst strong evidence has been provided that suggests that these shRNA are effective, no quantification of knockdown efficiency is presented. 1 Western blot is provided (with a single n of each shRNA), and representative immunolabelled neurons (for Mff but not for Fis1). Authors should provide quantification of knockdown efficiency for both Mff and Fis1 from immunolabelled neurons (i.e. electroporated vs non-electroporated from same field of view) to indicate efficiency of knockdown in their model system.

In main methods and results text, authors specify that 'a mixture of shRNA' is used for knockdown. In supplementary fig 1, 3x (triple) shRNA have been used in HEK cells/ Westerns, and 2x shRNA (539+665) have been used in neurons. Authors should clarify in main text which shRNA have been used for all experiments, and should quantify the efficiency of knockdown from that system (i.e. if 539+665 have been used throughout, 539+665 should be quantified).

We have addressed this concern by quantifying endogenous MFF protein levels in individual MFF shRNA electroporated neurons and nearby un-electroporated neurons. We observe a decrease of more than 65% in shRNA electroporated neurons. We have more carefully stated that all experiments (except the HEK cell validation) have been performed with shRNAs at a 1:1 ratio of the 539 and 665 constructs.

2) Related to above, Fis1 knockdown looks less efficient from the Western blot, and no evidence provided of knockdown efficiency in neurons. Therefore, relative lack of effect of Fis1 knockdown on mitochondrial length in Fig 2 may be due to less efficient knockdown.

As the Fis1 data was not essential to the major objective and findings in the paper, we have removed this data and claims associated with it from the manuscript.

3) More details are required regarding the assay in Fig 3. It is clear how this would be used to measure fusion; how does it measure fission?

We have included a better description of how we performed the assay in Figure 3. Essentially, we are likely under-sampling fission as we can only unambiguously distinguish fission events following a fusion event (i.e. when a “white” mitochondrion resulting from a fusion event between converted and unconverted mt-mEOS2 becomes two or more smaller white mitochondria). Without this method it is not possible to distinguish mitochondria simply overlapping or passing over each other from those that undergo fission. This new approach represents a significant improvement over previously published techniques.

4) In Fig 7, the peak height of pHluorin fluorescence represents an equilibrium between exocytosis and compensatory endocytosis occurring during stimulation. Thus, the difference between the two peaks may be driven by reduction in exocytosis, or by an increased rate of endocytosis during the stimulus train.

To confirm that this is due to a reduction in evoked exocytosis, this assay should be performed in the presence of a vATPase inhibitor (eg bafilomycin) to block vesicle reacidification and allow quantification of exocytosis specifically.

We appreciate this concern for more precise validation. A previous report showed that Mff knockdown neurons have reduced endo/exocytosis ratio with bafilomycin incubation (Fig. 5k-m from Li et al¹³). Although this study claimed that MFF plays role for endocytosis of synaptic vesicles with Drp1, the graph displays that the exocytosis level is lower in Mff-deficient presynaptic boutons with bafilomycin treatment. Also, our newly added data (**Fig. 7i**) demonstrates that total synaptic vesicle pool size is not significantly different between control and Mff knockdown axons. Therefore, decreased evoked presynaptic vesicle exocytosis is not due to reduced vesicle pool size.

5) In the model diagram, the authors propose that in terminals lacking mitochondria, there is an increased Ca²⁺ build-up, and increased neurotransmitter release (Fig. S7 C and D, left). In terminals with normal mitochondria, there is reduced neurotransmitter release (Fig. S7C right), and in terminals with large mitochondria, there is a strong reduction in neurotransmitter release (S7D right). This is an elegant proposal, but evidence for this is not presented in this paper. Instead, authors have demonstrated that in terminals with long mitochondria there is a reduction in peak height of fluorescence during stimulation, compared to terminals with normal mitochondria

(however this requires clarification – see minor comments). Moreover, several studies have shown that neurotransmitter release is similar in synapses with and without mitochondria (eg Pathak et al JBC 2015).

Authors could present evidence for their hypothesis, by comparing not only terminals with long/short mitochondria (control vs shMFF), but also comparing with/without mitochondria from within the same axon – this data will be available through the exocytosis assay for Fig 7.

We agree with this reviewer's concern. As suggested, we now present analysis where we compared presynaptic Ca^{2+} and SV exocytosis with/without elongated mitochondria in Mff-deficient neurons and these results demonstrate that mitochondria-free presynaptic bouton in Mff knockdown axons display higher evoked presynaptic Ca^{2+} and synaptic vesicle exocytosis than at presynaptic boutons with elongated mitochondria-associated ones. This strongly argues that the changes observed in presynaptic calcium dynamics and neurotransmitter release at boutons associated with elongated mitochondria in Mff knockdown axons are not due to a global or unspecific effects of Mff knockdown. We have added this new data in **Fig. S7b-c**. The previous study (Pathak et al., JBC, 2015) applied a very strong, non-physiological stimulation (600APs) and different imaging solution (without glucose) to earlier stage neurons (11DIV for rat neurons or 8-9DIV for mouse neurons) compared to our condition (mouse cortical PNs at 21DIV). These differences likely underlie the distinct results. Furthermore, our lab as well as the lab of Dr Kittler have recently published results demonstrating that along the same axon, presynaptic boutons associated with mitochondria display lower cytoplasmic Ca^{2+} accumulation and presynaptic vesicle release than boutons not associated with mitochondria during physiological regimes of evoked presynaptic release (20AP @ 10Hz for example; ^{7,14}).

Minor

Used throughout: vesicle release. Synaptic vesicles aren't 'released'. Should be neurotransmitter release, or synaptic vesicle exocytosis.

We apologize for this mistake and have corrected it throughout the manuscript.

Please clarify what is meant 'high density' cultures (line 110)

We have altered the text to state the actual number of cells plated upon dissociation; however, we cannot be certain the final number of cells per well as astrocytes are highly proliferative in these culture conditions.

Please clarify what is meant by 'control' in Fig 7F: is control non-knockdown neurons with small mitochondria? Or terminals without mitochondria?

Control means non-knockdown neurons with small mitochondria. We apologize for this confusion. We added more details in figure legends.

Please clarify throughout how many cultures/animals have been used. This is sometimes presented (eg. Fig 2 line 687) but has not been uniformly presented for all figures.

We have made sure to add this information into each figure legend, so it is clear for each figure.

REFERENCES CITED

- 1 Connolly, N. M. C. *et al.* Guidelines on experimental methods to assess mitochondrial dysfunction in cellular models of neurodegenerative diseases. *Cell Death Differ* **25**, 542-572, doi:10.1038/s41418-017-0020-4 (2018).
- 2 Williams, T., Courchet, J., Viollet, B., Brenman, J. E. & Polleux, F. AMP-activated protein kinase (AMPK) activity is not required for neuronal development but regulates axogenesis during metabolic stress. *Proc Natl Acad Sci U S A* **108**, 5849-5854, doi:10.1073/pnas.1013660108 (2011).
- 3 Mairet-Coello, G. *et al.* The CAMKK2-AMPK kinase pathway mediates the synaptotoxic effects of Abeta oligomers through Tau phosphorylation. *Neuron* **78**, 94-108, doi:10.1016/j.neuron.2013.02.003 (2013).
- 4 Rangaraju, V., Calloway, N. & Ryan, T. A. Activity-driven local ATP synthesis is required for synaptic function. *Cell* **156**, 825-835, doi:10.1016/j.cell.2013.12.042 (2014).
- 5 Ashrafi, G., Wu, Z., Farrell, R. J. & Ryan, T. A. GLUT4 Mobilization Supports Energetic Demands of Active Synapses. *Neuron* **93**, 606-615 e603, doi:10.1016/j.neuron.2016.12.020 (2017).
- 6 Jang, S. *et al.* Glycolytic Enzymes Localize to Synapses under Energy Stress to Support Synaptic Function. *Neuron* **90**, 278-291, doi:10.1016/j.neuron.2016.03.011 (2016).
- 7 Kwon, S. K. *et al.* LKB1 Regulates Mitochondria-Dependent Presynaptic Calcium Clearance and Neurotransmitter Release Properties at Excitatory Synapses along Cortical Axons. *PLoS Biol* **14**, e1002516, doi:10.1371/journal.pbio.1002516 (2016).
- 8 Verstreken, P. *et al.* Synaptic mitochondria are critical for mobilization of reserve pool vesicles at Drosophila neuromuscular junctions. *Neuron* **47**, 365-378, doi:10.1016/j.neuron.2005.06.018 (2005).
- 9 Breckwoldt, M. O. *et al.* Multiparametric optical analysis of mitochondrial redox signals during neuronal physiology and pathology in vivo. *Nat Med* **20**, 555-560, doi:10.1038/nm.3520 (2014).
- 10 Pathak, D. *et al.* The role of mitochondrially derived ATP in synaptic vesicle recycling. *J Biol Chem* **290**, 22325-22336, doi:10.1074/jbc.M115.656405 (2015).
- 11 Suzuki, R., Hotta, K. & Oka, K. Transitional correlation between inner-membrane potential and ATP levels of neuronal mitochondria. *Sci Rep* **8**, 2993, doi:10.1038/s41598-018-21109-2 (2018).
- 12 Cagalinec, M. *et al.* Role of Mitochondrial Dynamics in Neuronal Development: Mechanism for Wolfram Syndrome. *PLoS Biol* **14**, e1002511, doi:10.1371/journal.pbio.1002511 (2016).
- 13 Li, H. *et al.* A Bcl-xL-Drp1 complex regulates synaptic vesicle membrane dynamics during endocytosis. *Nat Cell Biol* **15**, 773-785, doi:10.1038/ncb2791 (2013).
- 14 Vaccaro, V., Devine, M. J., Higgs, N. F. & Kittler, J. T. Miro1-dependent mitochondrial positioning drives the rescaling of presynaptic Ca²⁺ signals during homeostatic plasticity. *EMBO Rep* **18**, 231-240, doi:10.15252/embr.201642710 (2017).

REVIEWERS' COMMENTS:

Reviewer #1 (Remarks to the Author):

The authors have adequately addressed all my previous concerns.

Reviewer #2 (Remarks to the Author):

The authors have successfully answered all my questions. They may want to consider transferring some of the new data currently in Fig S7 to the main figures but I will leave that to their discretion. I acknowledge the 'tour the force' in analysing calcium levels and synaptic vesicle fusion and individual boutons, a job well done!

I have no further reservations concerning publication of this manuscript.

Reviewer #3 (Remarks to the Author):

The authors have addressed my comments, however minor additions to the paper are important for interpretation of analysis. This paper provides important insight into how mitochondria affect synaptic biology.

Minor comments:

The addition of raw fluorescence (Fmax) values for 7e and 7i is only meaningful if the acquisition conditions were identical for each n and between conditions (eg same laser intensity, same exposure time etc). Please specify in methods/figure legend that this was the case.

The authors have provided evidence of quantification of knockdown efficiency. However, given the importance of this data for the rest of the paper, the authors should note what n has been used for this analysis. Are individual data points individual ROIs from a single coverslip? Multiple coverslips? Multiple preps?

REVIEWERS' COMMENTS:

Reviewer #1 (Remarks to the Author):

The authors have adequately addressed all my previous concerns.

We thank the reviewer for their enthusiasm towards our manuscript and for their suggestions which allowed us to improve it.

Reviewer #2 (Remarks to the Author):

The authors have successfully answered all my questions. They may want to consider transferring some of the new data currently in Fig S7 to the main figures but I will leave that to their discretion. I acknowledge the 'tour the force' in analyzing calcium levels and synaptic vesicle fusion and individual boutons, a job well done!
I have no further reservations concerning publication of this manuscript.

We thank the reviewer for their enthusiasm towards our manuscript and for their suggestions which allowed us to improve it!

We believe the new data presented in Supplementary Figure 7 is important but believe it is best presented in the supplemental section of the work.

Reviewer #3 (Remarks to the Author):

The authors have addressed my comments, however minor additions to the paper are important for interpretation of analysis. This paper provides important insight into how mitochondria affect synaptic biology.

We thank the reviewer for their enthusiasm towards our manuscript and for their suggestions which allowed us to improve it.

Minor comments:

The addition of raw fluorescence (F_{max}) values for $7e$ and $7i$ is only meaningful if the acquisition conditions were identical for each n and between conditions (eg same laser intensity, same exposure time etc). Please specify in methods/figure legend that this was the case.

We agree with the reviewer, and apologize for the omission of this information. We have addressed this by adding the requested information in the figure legend for Figure 7.

The authors have provided evidence of quantification of knockdown efficiency. However, given the importance of this data for the rest of the paper, the authors should

note what n has been used for this analysis. Are individual data points individual ROIs from a single coverslip? Multiple coverslips? Multiple preps?

We agree with this comment as well, and apologize for this omission. The results were obtained from two independent cultures, which were independently stained via immunocytochemistry. Imaging was performed with the same exact settings for all acquisitions as intensities were compared. We have addressed this by adding the above information into the figure legend for Supplementary Figure 1.